# SwinGNN: Rethinking Permutation Invariance in Diffusion Models for Graph Generation

**Qi Yan**                                                                     *qi.yan@ece.ubc.ca*
*University of British Columbia*
*Vector Institute for AI*

**Zhengyang Liang**                                                            *marxas@tongji.edu.cn*
*Tongji University*

**Yang Song**[*]                                                              *songyang@openai.com*
*OpenAI*

**Renjie Liao**                                                                *rjliao@ece.ubc.ca*
*University of British Columbia*
*Vector Institute for AI*
*Canada CIFAR AI Chair*

**Lele Wang**                                                                  *lelewang@ece.ubc.ca*
*University of British Columbia*

**Reviewed on OpenReview:** *https://openreview.net/forum?id=abfi5plvQ4*

## Abstract

Permutation-invariant diffusion models of graphs achieve the invariant sampling and invariant loss functions by restricting architecture designs, which often sacrifice empirical performances. In this work, we first show that the performance degradation may also be contributed by the increasing modes of target distributions brought by invariant architectures since 1) the optimal one-step denoising scores are score functions of Gaussian mixtures models (GMMs) whose components center on these modes and 2) learning the scores of GMMs with more components is often harder. Motivated by the analysis, we propose *SwinGNN* along with a simple yet provable trick that enables permutation-invariant sampling. It benefits from more flexible (non-invariant) architecture designs and permutation-invariant sampling. We further design an efficient 2-WL message passing network using the shifted-window self-attention. Extensive experiments on synthetic and real-world protein and molecule datasets show that SwinGNN outperforms existing methods by a substantial margin on most metrics. Our code is released at `https://github.com/qiyan98/SwinGNN`.

## 1 INTRODUCTION

Diffusion models (Sohl-Dickstein et al., 2015; Ho et al., 2020; Song et al., 2021a) have recently emerged as a powerful class of deep generative models. They can generate high-dimensional data, *e.g.*, images and videos (Dhariwal & Nichol, 2021; Ho et al., 2022), at unprecedented high qualities. While originally designed for continuous data, diffusion models have inspired new models for graph generation that involves discrete data and has a wide range of applications, including molecule generation (Jin et al., 2018), code completion (Brockschmidt et al., 2019), urban planning (Chu et al., 2019), scene graph generation (Suhail et al., 2021), and neural architecture search (Li et al., 2022).

---

[*]Work done at Stanford.

There exist two ways to generalize diffusion models to graphs. The first one is simply treating adjacency matrices as images and applying existing techniques built for continuous data. The only additional challenge compared to image generation is that a desirable probability distribution over graphs should be invariant to the permutation of nodes. Niu et al. (2020); Jo et al. (2022) construct permutation equivariant score-based models that induce permutation invariant distributions. Post-hoc thresholding is needed to convert sampled continuous adjacency matrices to binary ones. The other one relies on the recently proposed discrete diffusion models (Austin et al., 2021) that naturally operate on binary adjacency matrices with discrete transitions. Vignac et al. (2023) shows that such models learn permutation invariant graph distributions by construction and achieve impressive results on molecule generation.

In this paper, we first clarify two key concepts of permutation invariance: invariant training losses and invariant sampling. For practical applications, graph generative models should ideally have an invariant sampling process, ensuring generated graphs that are isomorphic have equal likelihoods regardless of node orders. Previous works achieve this goal by using invariant diffusion models trained with invariant losses. However, such invariant models often achieve worse empirical performances than their non-invariant counterparts. This is likely due to 1) invariant models have more restrictive architecture designs and 2) their target distributions having more modes. Specifically, since the optimal one-step denoising scores are score functions of Gaussian mixtures models (GMMs) whose components center on these modes, learning the scores of GMMs with more components is often harder. Importantly, while an invariant loss is sufficient, it is not necessary for invariant sampling. We present a simple technique of randomly permuting generated graphs that provably enables any graph generative model to achieve permutation-invariant sampling.

Motivated by the analysis, we propose a non-invariant diffusion model, called SwinGNN, to embrace more powerful architecture designs while still maintaining permutation-invariant sampling. Inspired by the expressive 2-WL graph neural networks (GNNs) (Morris et al., 2021) and SwinTransformers (Liu et al., 2021), our SwinGNN performs efficient edge-to-edge 2-WL message passing via shifted window based self-attention mechanism and customized graph downsampling/upsampling operators, thus being scalable to generate large graphs (*e.g.*, over 500 nodes). Note that directly applying 2-WL GNNs would have significantly higher computational costs whereas directly applying SwinTransformers would lead to worse performances since they are essentially 1-WL GNNs. Our model aligns with the rationale of $k$-order GNN (Maron et al., 2019b) or $k$-WL GNN (Morris et al., 2019), enhancing network expressivity by utilizing $k$-tuples of nodes ($k = 2$ in our case). Further, we thoroughly investigate the recent advances of diffusion models for image (Karras et al., 2022; Song & Ermon, 2020) and identify several techniques that significantly improve the sample quality of graph generation. Extensive experiments on synthetic and real-world molecule and protein datasets show that our SwinGNN achieves state-of-the-art performances, surpassing the existing models by several orders of magnitude in most metrics.

## 2 RELATED WORK

Generative models of graphs (*a.k.a.*, random graph models) have been studied in mathematics, network science, and other subjects for decades since the seminal Erdős–Rényi model (Erdős & Rényi, 1959). Most of these models, *e.g.*, Watts–Strogatz model (Watts & Strogatz, 1998) and Barabási–Albert model (Albert & Barabási, 2002), are mathematically tractable, *i.e.*, one can rigorously analyze their properties such as degree distributions. In the context of deep learning, deep generative models for graphs (Liao, 2022) prioritize fitting complex distributions of real-world graphs over obtaining tractable properties and have achieved impressive performances. They can be broadly classified based on the generative modeling techniques.

The first class of methods relies on diffusion models (Sohl-Dickstein et al., 2015; Ho et al., 2020) that achieve great successes in image and video generation. We focus on continuous diffusion graph generative models which treat the adjacency matrices as images. (Niu et al., 2020) proposes a permutation-invariant score-matching objective along with a GNN architecture for generating binary adjacency matrices by thresholding continuous matrices. Jo et al. (2022) extends this approach to handle node and edge attributes via stochastic differential equation framework (Song et al., 2021b). Following this line of research, we investigate the limiting factors of these models and propose our improvements that achieve state-of-the-art performances.

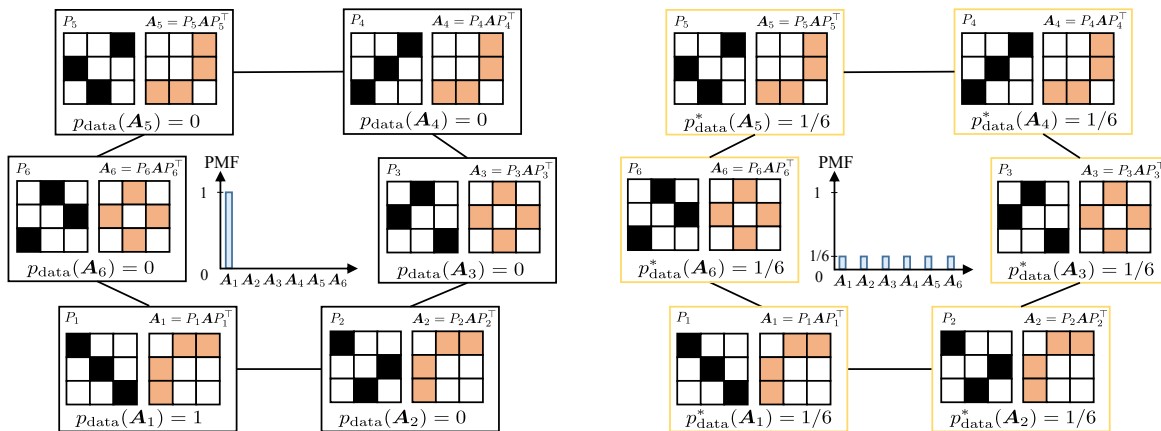

Figure 1: Data distribution and target distribution for a 3-node tree graph. For permutation matrix $\boldsymbol{P}_i$ and adjacency matrix $\boldsymbol{A}_i$, filled/blank cells mean one/zero. The probability mass function (PMF) highlights the difference in modes. Our example also shows graph automorphism (*e.g.*, $\boldsymbol{P}_1$ and $\boldsymbol{P}_2$).

Besides continuous diffusion, discrete diffusion based graph generative models also emerged recently. Vignac et al. (2023) proposes a permutation-invariant model based on discrete diffusion models (Austin et al., 2021; Hoogeboom et al., 2021; Johnson et al., 2021). For the discrete graph data, *e.g.*, binary adjacency matrices or categorical node and edge attributes, discrete diffusion is more intuitive than the continuous counterpart.

Apart from the above diffusion based models, there also exist models based on generative adversarial networks (GANs) (De Cao & Kipf, 2018; Krawczuk et al., 2021; Martinkus et al., 2022), variational auto-encoders (VAEs) (Kipf & Welling, 2016; Simonovsky & Komodakis, 2018; Jin et al., 2018; Vignac & Frossard, 2022), normalizing flows (Liu et al., 2019; Madhawa et al., 2019; Lippe & Gavves, 2021; Luo et al., 2021), and autoregressive models (You et al., 2018; Liao et al., 2019; Mercado et al., 2021). Among them, autoregressive based models enjoy the best empirical performance, although they are not invariant to permutations.

Recently, Han et al. (2023) propose modeling the graph distribution by summing over the isomorphism class of adjacency matrices with autoregressive models, where a graph $\mathcal{G}$ with an adjacency matrix $\boldsymbol{A}$ of $n$ nodes admits a probability of

$$p(\mathcal{G}) = \sum_{\boldsymbol{A}_i \in \mathcal{I}_{\boldsymbol{A}}} p(\boldsymbol{A}_i), \tag{1}$$

where $\mathcal{I}_{\boldsymbol{A}}$ is the isomorphism class of $\boldsymbol{A}$ with a size up to $O(n!)$.

## 3 PRELIMINARIES

**Notation.** A graph $\mathcal{G} = (\mathcal{V}, \mathcal{E})$ comprises a node set $\mathcal{V}$ with $n$ nodes $\mathcal{V} = \{1, 2, \ldots, n\}$ and an edge set $\mathcal{E} \subseteq \mathcal{V} \times \mathcal{V}$. We focus on simple graphs, *i.e.*, unweighted and undirected graphs without self-loops or multiple edges[1]. Such a graph can be represented by an adjacency matrix $\boldsymbol{A}^\pi \in \{0, 1\}^{n \times n}$, where node ordering $\pi$ implies the matrix's row/column order. A permutation matrix $\boldsymbol{P}_{\pi_1 \to \pi_2}$ denotes a bijection between two node orderings $\pi_1$, $\pi_2$, *i.e.*, $\boldsymbol{A}^{\pi_2} = \boldsymbol{P}_{\pi_1 \to \pi_2} \boldsymbol{A}^{\pi_1} \boldsymbol{P}_{\pi_1 \to \pi_2}^\top$. We omit the node ordering notations for simplicity. Let $\mathcal{S}_n$ be all $n!$ permutation matrices of $n$ nodes. Any $\boldsymbol{P} \in \mathcal{S}_n$ that maps $\boldsymbol{A}$ to itself, *i.e.*, $\boldsymbol{A} = \boldsymbol{P}\boldsymbol{A}\boldsymbol{P}^\top$ denotes a *graph automorphism* of $\boldsymbol{A}$. Two graphs $\mathcal{G}_1, \mathcal{G}_2$ with adjacency matrices $\boldsymbol{A}_1, \boldsymbol{A}_2$ are *isomorphic* if and only if there exists $\boldsymbol{P} \in \mathcal{S}_n$ such that $\boldsymbol{P}\boldsymbol{A}_1\boldsymbol{P}^\top = \boldsymbol{A}_2$. The *isomorphism class* of $\boldsymbol{A}$, denoted by $\mathcal{I}_{\boldsymbol{A}} := \{\boldsymbol{P}\boldsymbol{A}\boldsymbol{P}^\top | \boldsymbol{P} \in \mathcal{S}_n\}$, is the set of adjacency matrices isomorphic to $\boldsymbol{A}$. We observe *i.i.d.* samples, $\mathcal{A} := \{\boldsymbol{A}_i\}_{i=1}^m \sim p_{\text{data}}(\boldsymbol{A})$, drawn from the unknown data distribution of graphs $p_{\text{data}}(\boldsymbol{A})$.

---

[1]Weighted graphs (*i.e.*, real-valued adjacency matrices) are easier to handle for continuous diffusion models since the thresholding step is unnecessary. Our model also extends to multigraphs with node and edge attributes (*e.g.*, molecules), detailed further in the experiment section and App. B.3.

Graph generative models aim to learn a distribution $p_\theta(\boldsymbol{A})$ that closely approximates $p_{\text{data}}(\boldsymbol{A})$. **Denoising Diffusion Models.** A denoising diffusion model consists of two parts: (1) a forward continuous-state Markov chain that gradually adds noise to observed data until it becomes standard normal noise and (2) a backward continuous-state Markov chain (learnable) that gradually denoises from the standard normal noise until it becomes observed data. The transition probability of the backward chain is typically parameterized by a deep neural network. Two consecutive transitions in the forward (backward) chain correspond to two noise levels that are increasing (decreasing). One can have discrete-time (finite noise levels) (Ho et al., 2020) or continuous-time (infinite noise levels) (Song et al., 2021b) denoising diffusion models.

In the context of graphs, if we treat an adjacency matrix $\boldsymbol{A}$ as an image, it is straightforward to apply these models. In particular, considering one noise level $\sigma$, the loss of denoising diffusion models is

$$\mathbb{E}_{p_{\text{data}}(\boldsymbol{A})p_\sigma(\tilde{\boldsymbol{A}}|\boldsymbol{A})} \left[ \|D_\theta(\tilde{\boldsymbol{A}}, \sigma) - \boldsymbol{A}\|_F^2 \right], \tag{2}$$

where $D_\theta(\tilde{\boldsymbol{A}}, \sigma)$ is the denoising network, $\tilde{\boldsymbol{A}}$ is the noisy data (graph), and $\| \cdot \|_F$ is the Frobenius norm. The forward transition probability is a Gaussian distribution $p_\sigma(\tilde{\boldsymbol{A}}|\boldsymbol{A}) \coloneqq \mathcal{N}(\tilde{\boldsymbol{A}}; \boldsymbol{A}, \sigma^2 \boldsymbol{I})$. Note that we add *i.i.d.* element-wise Gaussian noise to the adjacency matrix. Based on the Tweedie's formula (Efron, 2011), one can derive the optimal denoiser $D_{\theta^*}(\tilde{\boldsymbol{A}}, \sigma) = \tilde{\boldsymbol{A}} + \sigma^2 \nabla_{\tilde{\boldsymbol{A}}} \log p_\sigma(\tilde{\boldsymbol{A}})$, where the *noisy data* distribution $p_\sigma(\tilde{\boldsymbol{A}}) \coloneqq \int p_{\text{data}}(\boldsymbol{A}) p_\sigma(\tilde{\boldsymbol{A}}|\boldsymbol{A}) d\boldsymbol{A}$ appears.

**Score-based Models.** Score-based models aim to learn the score function (the gradient of the log density w.r.t. data) of the data distribution $p_{\text{data}}(\boldsymbol{A})$, denoted by $s(\boldsymbol{A}) \coloneqq \nabla_{\boldsymbol{A}} \log p_{\text{data}}(\boldsymbol{A})$. Since $p_{\text{data}}(\boldsymbol{A})$ is unknown, one needs to leverage techniques such as denoising score matching (DSM) (Vincent, 2011) to train a score estimation network $s_\theta$. Similar to diffusion models, we add *i.i.d.* element-wise Gaussian noise to data, *i.e.*, $p_\sigma(\tilde{\boldsymbol{A}}|\boldsymbol{A}) = \mathcal{N}(\tilde{\boldsymbol{A}}; \boldsymbol{A}, \sigma^2 \boldsymbol{I})$. For a single noise level $\sigma$, the DSM loss is

$$\mathbb{E}_{p_{\text{data}}(\boldsymbol{A})p_\sigma(\tilde{\boldsymbol{A}}|\boldsymbol{A})} \left[ \|s_\theta(\tilde{\boldsymbol{A}}, \sigma) - \nabla_{\tilde{\boldsymbol{A}}} \log p_\sigma(\tilde{\boldsymbol{A}}|\boldsymbol{A})\|_F^2 \right], \tag{3}$$

where $\nabla_{\tilde{\boldsymbol{A}}} \log p_\sigma(\tilde{\boldsymbol{A}}|\boldsymbol{A}) = \frac{\boldsymbol{A} - \tilde{\boldsymbol{A}}}{\sigma^2}$. Minimizing Eq. (3) almost surely leads to an optimal score network $s_{\theta^*}(\tilde{\boldsymbol{A}}, \sigma)$ that matches the score of the noisy data distribution, *i.e.*, $s_{\theta^*}(\tilde{\boldsymbol{A}}, \sigma) = \nabla_{\tilde{\boldsymbol{A}}} \log p_\sigma(\tilde{\boldsymbol{A}})$ (Vincent, 2011). Further, the denoising diffusion models and score-based models are essentially the same (Song et al., 2021b). The optimal denoiser of Eq. (2) and the optimal score estimator of Eq. (3) are inherently connected by $D_{\theta^*}(\tilde{\boldsymbol{A}}, \sigma) = \tilde{\boldsymbol{A}} + \sigma^2 s_{\theta^*}(\tilde{\boldsymbol{A}}, \sigma)$. We use both terms interchangeably in what follows.

**DSM Estimates the Score of GMMs.** Our training set consists of *i.i.d.* samples (adjacency matrices) $\{\boldsymbol{A}_i\}_{i=1}^m$. The corresponding empirical data distribution[2] is a mixture of Dirac delta distributions, *i.e.*, $p_{\text{data}}(\boldsymbol{A}) \coloneqq \frac{1}{m} \sum_{i=1}^m \delta(\boldsymbol{A} - \boldsymbol{A}_i)$, from which we can get the closed-form of the empirical noisy data distribution $p_\sigma(\tilde{\boldsymbol{A}}) \coloneqq \frac{1}{m} \sum_{i=1}^m \mathcal{N}(\tilde{\boldsymbol{A}}; \boldsymbol{A}_i, \sigma^2 \boldsymbol{I})$. $p_\sigma(\tilde{\boldsymbol{A}})$ is an $m$-component GMM with uniform weights, and the DSM objective in Eq. (3) learns its score function.

## 4 PERMUTATION INVARIANT LOSS AND SAMPLING

Existing models employ permutation invariant losses to attain permutation invariant sampling, a feature essential for real-world applications. However, we demonstrate that permutation invariant losses lead to degraded empirical performances. Alternatively, we introduce a simple yet provable technique to ensure invariant sampling without requiring an invariant loss, fostering flexibility in model architecture design and inspiring our non-invariant method.

### 4.1 Challenges of Permutation Invariant Loss

**Theoretical Analysis.** As shown in Fig. 1, the empirical graph distribution $p_{\text{data}}(\boldsymbol{A})$ may only assign a non-zero probability to a single observed adjacency matrix in its isomorphism class. The ultimate goal of a graph generative model is to match this empirical distribution, which may be biased by the observed

---

[2]With a slight abuse of notation, we refer to both the data distribution and its empirical version as $p_{\text{data}}$ since the data distribution is unknown and will not be often used.

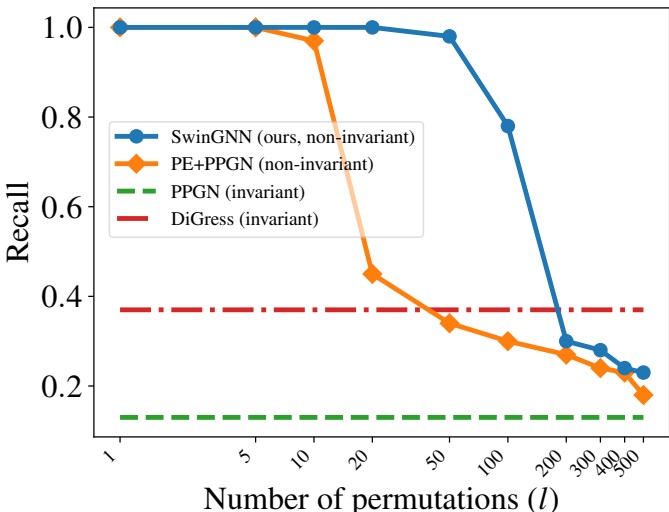

Figure 2: Invariant models perform significantly worse than non-invariant models when the number of applied permutations ($l$) is small.

permutation. However, the target distribution that generative models are trained to match may differ from the empirical one, dependent on the model design w.r.t. permutation symmetry. For clarity, we define the *effective target distribution* as the closest distribution (*e.g.*, measured in total variation distance) to the empirical data distribution achievable by the generative model, assuming sufficient data and model capacity.

Previous works (Niu et al., 2020; Xu et al., 2022; Hoogeboom et al., 2022) learn permutation invariant models $p_\theta(\boldsymbol{A})$ with permutation invariant losses via equivariant networks. We argue the induced invariant effective target distributions are hard to learn. Let the training set only contain one graph $\boldsymbol{A}_1$, *e.g.*, in Fig. 1. Even if we optimize the invariant model distribution $p_\theta(\boldsymbol{A})$ towards the empirical one $p_{\text{data}}(\boldsymbol{A})$ to the optimum, they can never exactly match. This is because if it does (*i.e.*, $p_{\text{data}}(\boldsymbol{A} = \boldsymbol{A}_1) = p_\theta(\boldsymbol{A} = \boldsymbol{A}_1) = 1$), $p_\theta(\boldsymbol{A})$ will also assign the same probability to isomorphic graphs of $\boldsymbol{A}_1$ due to permutation invariance, thus violating the probability sum-to-one rule. Instead, the optimal $p_\theta(\boldsymbol{A})$ (*i.e.*, the effective target distribution) will assign equal probability $1/|\mathcal{I}_{\boldsymbol{A}_1}|$ to isomorphic graphs of $\boldsymbol{A}_1$.

Formally, given a training set of adjacency matrices $\{\boldsymbol{A}_i\}_{i=1}^m$, one can construct the union set of each graph's isomorphism class, denoted as $\mathcal{A}^* = \bigcup_{i=1}^m \mathcal{I}_{\boldsymbol{A}_i}$. The corresponding mixture distribution is $p_{\text{data}}^*(\boldsymbol{A}) := \frac{1}{Z} \sum_{\boldsymbol{A}^* \in \mathcal{A}^*} \delta(\boldsymbol{A} - \boldsymbol{A}^*)$, where $Z = |\mathcal{A}^*| = O(n!m)$ is the normalizing constant. Note that $Z = n!m$ may not be always achievable due to graph automorphism.

**Lemma 4.1.** *Assume at least one training graph has $\Omega(n!)$ distinct adjacency matrices in its isomorphism class. Let $\mathcal{P}$ denote all discrete permutation invariant distributions. The closest distributions in $\mathcal{P}$ to $p_{\text{data}}$, measured by total variation, have at least $\Omega(n!)$ modes. If, in addition, we restrict $\mathcal{P}$ to be the set of permutation invariant distributions such that $p(\boldsymbol{A}_i) = p(\boldsymbol{A}_j) > 0$ for all matrices in the training set $\{\boldsymbol{A}_l\}_{l=1}^m$, then the closest distribution is given by $\arg\min_{q \in \mathcal{P}} TV(q, p_{\text{data}}) = p_{\text{data}}^*$.*

Under mild conditions, $p_{\text{data}}^*(\boldsymbol{A})$ of $O(n!m)$ modes becomes the effective target distribution, which is the case of prior invariant models using equivariant networks for invariant losses (see App. A for details). In contrast, if we employ a non-equivariant network (*i.e.*, the underlying training objective is not invariant), the effective target distribution would be $p_{\text{data}}(\boldsymbol{A})$ which only has $O(m)$ modes. Moreover, the modes of the Dirac delta target distributions dictate the components of the GMMs in the diffusion models, with each component precisely centered on a target mode. With the invariant loss, the GMMs have the form of $p_\sigma^*(\tilde{\boldsymbol{A}}) := \frac{1}{Z} \sum_{\boldsymbol{A}_i^* \in \mathcal{A}^*} \mathcal{N}(\tilde{\boldsymbol{A}}; \boldsymbol{A}_i, \sigma^2 \boldsymbol{I})$ with an $O(n!)$ factor more components than the non-invariant loss (detailed in App. A.3). Arguably, learning with a permutation invariant loss is much harder than a non-invariant one, implied by the $O(n!)$-factor surge in modes of target distribution and components of GMMs.

**Empirical Investigation.** We typically observe one adjacency matrix from its isomorphism class in training data $\{\boldsymbol{A}_i\}_{i=1}^m$. By applying permutation $n!$ times, one can construct $p_{\text{data}}^*$ from $p_{\text{data}}$. We define a trade-off between them, called the $l$-permuted empirical distribution: $p_{\text{data}}^l(\boldsymbol{A}) := \frac{1}{ml} \sum_{i=1}^m \sum_{j=1}^l \delta(\boldsymbol{A} - \boldsymbol{P}_j \boldsymbol{A}_i \boldsymbol{P}_j^\top)$, where $\boldsymbol{P}_1, \ldots, \boldsymbol{P}_l$ are $l$ distinct permutation matrices. $p_{\text{data}}^l$ has $O(lm)$ modes governed by $l$. With proper permutation matrices, $p_{\text{data}}^l = p_{\text{data}}$ when $l = 1$ and $p_{\text{data}}^l \approx p_{\text{data}}^*$ [3] when $l = n!$. To study the impact of the mode count in the effective target distribution on empirical performance, we use $p_{\text{data}}^l$ as the diffusion model's target by tuning $l$. An equivariant network matches the invariant target $p_{\text{data}}^*$, having $O(n!m)$ modes when $l = n!$. For $l < n!$, one must use a non-equivariant network to learn a non-invariant training objective.

To assess empirical performance, we conduct experiments on a toy dataset of 10 random regular graphs, each with 16 nodes and degrees in $[2, 11]$. The value of $l$ ranges from 1 to 500 and we ensure training is converged for all models. We use two invariant models as baselines for learning $p_{\text{data}}^*$: DiGress (Vignac et al., 2023) and PPGN (Maron et al., 2019a), with PPGN being notably expressive as a 3WL-discriminative GNN. For non-invariant models matching $p_{\text{data}}^l$ with $l < n!$, we add index-based positional embedding to PPGN and compare it with our SwinGNN model (detailed in Sec. 5.1). Please see App. B.1 for more details.

We use *recall* as the metric, which is defined by the proportion of generated graphs that are isomorphic to any training graph and ranges between 0 and 1. The recall requires isomorphism testing and is invariant to permutation, which is fair for all models. A higher recall indicates a strong capacity to capture the toy data distribution. In Fig. 2, invariant models, regardless of using discrete (DiGress) or continuous (PPGN) Markov chains in the diffusion processes, fail to achieve high recall. Conversely, non-invariant models perform exceptionally well when $l$ is small and the target distributions have modest imposed permutations. As $l$ goes up, the sample quality of non-invariant models drops significantly, indicating the learning difficulties incurred by more modes [4]. Notably, in practical applications of non-invariant models, one typically chooses $l = 1$, often resulting in empirically stronger performance compared to invariant counterparts.

## 4.2 Technique to Reclaim Invariant Sampling

While non-invariant models show stronger performances empirically, they cannot guarantee permutation-invariant sampling. We present the following lemma to restore the invariant sampling.

**Lemma 4.2.** *Let $\boldsymbol{A}$ be a random adjacency matrix distributed according to any graph distribution on $n$ vertices. Let $\boldsymbol{P}_r \sim \text{Unif}(\mathcal{S}_n)$ be uniform over the set of permutation matrices. Then, the induced distribution of the random matrix $\boldsymbol{A}_r = \boldsymbol{P}_r \boldsymbol{A} \boldsymbol{P}_r^\top$, denoted as $q_\theta(\boldsymbol{A}_r)$, is permutation invariant, i.e., $q_\theta(\boldsymbol{A}_r) = q_\theta(\boldsymbol{P} \boldsymbol{A}_r \boldsymbol{P}^\top), \forall \boldsymbol{P} \in \mathcal{S}_n$.*

This trick is applicable to all types of generative models. Note that the random permutation does not go beyond the isomorphism class. Although $q_\theta$ is provably invariant, it captures the same amount of isomorphism classes as $p_\theta$. Graphs generated from $q_\theta$ must have isomorphic counterparts that $p_\theta$ could generate. Thus far, we have uncovered two insights into permutation invariance in graph generative models: 1) invariant loss ensures invariant sampling but may impair empirical performances, and 2) invariant loss is not a prerequisite for invariant sampling. These insights prompt rethinking the prevailing design of invariant models using invariant losses. Aiming to accurately capture the complex real-world graph distributions while preserving sampling invariance, we introduce a novel diffusion model, integrating non-invariant loss with the invariant sampling technique in Lemma 4.2.

## 5 METHOD

### 5.1 Efficient High-order Graph Transformer

A continuous denoising network for graph data takes a noisy adjacency matrix $\tilde{\boldsymbol{A}}$ and its noise level $\sigma$ as input and outputs a denoised adjacency matrix. However, unlike in typical graph representation learning, our model carries out edge regression without clear graph topology (*i.e.*, no sparse binary adjacency matrices).

---

[3] They are identical without non-trivial automorphisms.
[4] Sample complexity analysis for non-invariant models is provided in App. A.4.

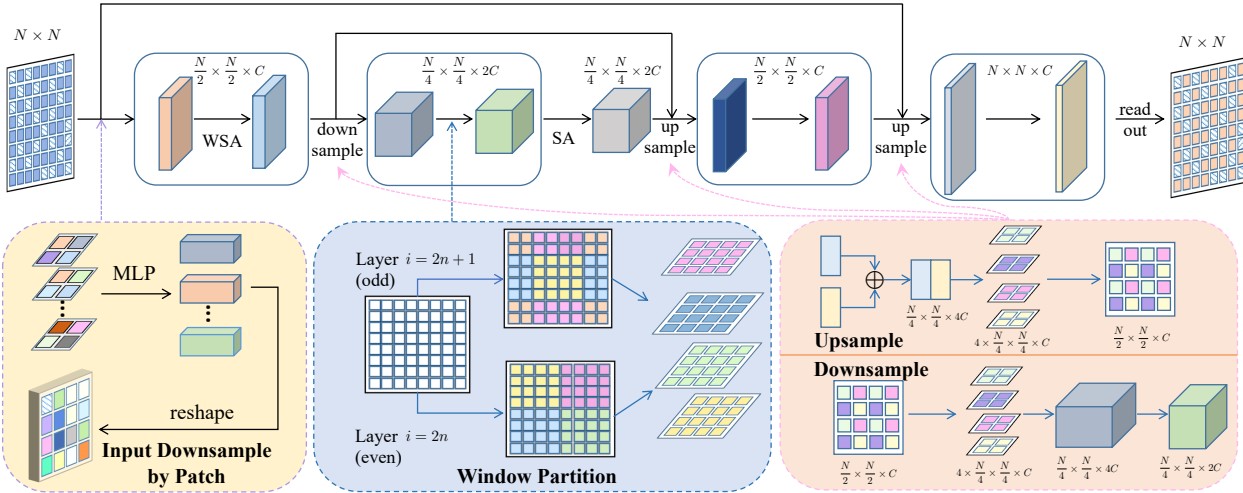

Figure 3: The overall architecture of our SwinGNN.

**Approximating 2-WL Message Passing.** The vanilla GNNs, such as the GCN (Kipf & Welling, 2017) have been shown to have the same expressiveness as the 1-dimensional Weisfeiler-Leman graph isomorphism heuristic (1-WL) in terms of distinguishing non-isomorphic graphs (Xu et al., 2019). Specifically, the graphs that the 1-WL algorithm fails to distinguish are also provably difficult to handle for classical GNNs. To address the limitations in expressiveness and performance, higher-order GNNs (Maron et al., 2019b; Morris et al., 2019) have been proposed to match the theoretical capacity of the higher-dimensional WL test, which is named $k$-WL GNNs in contrast to the vanilla GNNs of 1-WL expressiveness. To improve model capacity, we draw inspiration from the $k$-order GNN (Maron et al., 2019b) and the $k$-WL GNN (Morris et al., 2019) that improve network expressivity using $k$-tuples of nodes. The $k$-WL discrimination power characterizes the isomorphism testing ability, which proves equivalent to the function approximation capacity (Chen et al., 2019). However, applying any expressive network of $k \geq 2$ is challenging in practice due to its poor scalability. In our case, we view noisy input data $\tilde{A} \in \mathbb{R}^{n \times n}$ as weighted fully-connected graphs with $n$ nodes, which have $O(n^k)$ $k$-element subsets, leading to a message passing complexity of $O(n^{2k})$.

To address the computation challenges of $k$-WL network, we choose $k = 2$ to simplify the internal order and introduce a novel graph transformer to approximate the 2-WL (*i.e.*, edge-to-edge) message passing with efficient attention mechanism. Our model, as shown in Fig. 3, treats each edge value in the input matrix as a token, apply transformers with self-attention (Vaswani et al., 2017) to update the token representations, and output final edge tokens for denoising.

Specifically, we apply window-based partitioning to confine self-attention to local representations to improve efficiency. The model splits the $n \times n$ entries into local windows of size $M \times M$ in the grid map and computes self-attention within each window in parallel. With $M^2$ tokens per window and some reasonable $M$, the self-attention complexity reduces to $O(n^2 M^2)$, a significant decrease from the original $O(n^4)$ in 2-WL message passing. However, this approach limits inter-window message passing. To enhance cross-window interactions, we apply a shifted window technique (Liu et al., 2021), alternating between regular and shifted windows to approximate dense edge-to-edge interactions.

Although SwinGNN shares the same window attention with SwinTransformers (Liu et al., 2021), they are fundamentally different. SwinGNN is specially designed to approximate the computationally-expensive 2-WL network, while SwinTransformers can be viewed as 1-WL GNN applied to patch-level tokens. We conduct experiments in Tab. 1, Tab. 2 and App. B.6 to showcase our model's superior performances.

SwinTransformers are general-purpose feature extraction backbones for vision tasks. They need to be customized by connecting to a task-specific prediction head, which we implement using UperNet (Xiao et al., 2018) in our baseline experiments. However, we'd like to clarify that SwinTransformers simply treat graphs as image tensors and do not account for network expressiveness.

**Multi-scale Edge Representation Learning.** The window self-attention complexity $O(n^2 M^2)$ is dependent on $n^2$, which hinders scaling for large graphs. To further reduce memory footprint and better capture long-range interaction, we apply channel mixing-based downsampling and upsampling layers to construct hierarchical graph representations. As shown in Fig. 3, in the downsampling layer, we split the edge representation tensor into four half-sized tensors by index parity (odd or even) along rows and columns, make concatenation along channel dimension, and update tokens in the downsized tensors independently using an MLP. The upsampling stage is carried out likewise in the opposite direction, following the skip connection that concatenates same-size token tensors along channels.

Putting things together, we propose our model, called SwinGNN, that efficiently learns edge representations via approximating 2-WL message passing.

## 5.2 Training and Sampling with SDE

We construct the noisy data distribution through stochastic differential equation (SDE) modeling with a continuous time variable $t \in [0, T]$. Let noise $\sigma$ evolve with $t$ and the noisy distribution reads as:

$$p_{\sigma(t)}(\tilde{\boldsymbol{A}}) = \frac{1}{m} \sum_{i=1}^{m} \mathcal{N}(\tilde{\boldsymbol{A}}; \boldsymbol{A}_i, \sigma^2(t)\boldsymbol{I}), \boldsymbol{A}_i \in \mathcal{A}. \tag{4}$$

Namely, $p_{\sigma(0)}$ is the Dirac delta data distribution, and $p_{\sigma(T)}$ is close to the pure Gaussian noise (sampling prior distribution). The general forward and reverse process SDEs (Song et al., 2021b) are defined by:

$$d\boldsymbol{A}_{+} = f_d(\boldsymbol{A}, t)dt + g_d(t)d\boldsymbol{W}, \tag{5}$$

$$d\boldsymbol{A}_{-} = [f_d(\boldsymbol{A}, t)dt - g_d(t)^2 \nabla_{\boldsymbol{A}} \log p_t(\boldsymbol{A})]dt + g_d(t)d\boldsymbol{W}, \tag{6}$$

where the $d\boldsymbol{A}_{+}$ and $d\boldsymbol{A}_{-}$ denote forward and reverse processes respectively, $f_d(\boldsymbol{A}, t)$ and $g_d(t)$ are the drift and diffusion coefficients, and $\boldsymbol{W}$ is the standard Wiener process. We solve for the $f_d(\boldsymbol{A}, t)$ and $g_d(t)$ so that the noisy distribution $p_t$ in Eq. (6) satisfies Eq. (4), and the solution is given by

$$f(\boldsymbol{A}, t) = \boldsymbol{0}, g(t) = \sqrt{2\dot{\sigma}(t)\sigma(t)},$$

which can be found in Eq. (9) of Song et al. (2021b) or Eq. (6) of Karras et al. (2022).

We select the time-varying noise strength $\sigma(t)$ to be linear with $t$, *i.e.*, $\sigma(t) = t$, as in Nichol & Dhariwal (2021); Song et al. (2021a), which turns the SDEs into

$$d\boldsymbol{A}_{+} = \sqrt{2t}d\boldsymbol{W}, d\boldsymbol{A}_{-} = -2t\nabla_{\boldsymbol{A}} \log p_t(\boldsymbol{A})dt + \sqrt{2t}d\boldsymbol{W}. \tag{7}$$

Further, we adopt network preconditioning for improved training dynamics following the Elucidating Diffusion Model (EDM) (Karras et al., 2022). First, instead of training DSM in Eq. (3), we use its equivalent form in Eq. (2), and parameterize denoising function $D_\theta$ with noise dependent scaling: $D_\theta(\tilde{\boldsymbol{A}}, \sigma) = c_s(\sigma)\tilde{\boldsymbol{A}} + c_o(\sigma)F_\theta(c_i(\sigma)\tilde{\boldsymbol{A}}, c_n(\sigma))$, where $F_\theta$ is the actual neural network and the other coefficients are summarized in App. B.4. In implementation, $D_\theta$ is a wrapper with preconditioning operations, and we construct $F_\theta$ using our SwinGNN model in Sec. 5.1. Second, we sample $\sigma$ with $\ln(\sigma) \sim \mathcal{N}(P_{\text{mean}}, P_{\text{std}}^2)$ to select noise stochastically in a broad range to draw training samples. Third, we apply the weighting coefficients $\lambda(\sigma) = 1/c_o(\sigma)^2$ on the denoising loss to improve training stability. The overall training objective is

$$\mathbb{E}_{\sigma, \boldsymbol{A}, \tilde{\boldsymbol{A}}} \left[ \lambda(\sigma) \| c_s(\sigma)\tilde{\boldsymbol{A}} + c_o(\sigma)F_\theta(c_i(\sigma)\tilde{\boldsymbol{A}}, c_n(\sigma)) - \boldsymbol{A} \|_F^2 \right]. \tag{8}$$

The target of $F_\theta$ is $\frac{\boldsymbol{A} - c_s(\sigma)\tilde{\boldsymbol{A}}}{c_o(\sigma)}$, an interpolation between pure Gaussian noise $\boldsymbol{A} - \tilde{\boldsymbol{A}}$ (when $\sigma \to 0$) and clean sample $\boldsymbol{A}$ (when $\sigma \to \infty$), downscaled by $c_o(\sigma)$. These measures altogether ease the training of $F_\theta$ by making the network inputs and targets have unit variance.

**Self-Conditioning.** We apply self-conditioning (Chen et al., 2023) to let the diffusion model rely on sample created by itself. Let $\hat{\boldsymbol{A}}$ denote the sample created by the denoiser function $D_\theta$ during the reverse process,

which is initialized as zero tensors and becomes available after the first sampling step. We have modified the denoising function to be $D_\theta(\widetilde{\boldsymbol{A}}, \hat{\boldsymbol{A}}, \sigma)$, allowing it to take the previously generated sample $\hat{\boldsymbol{A}}$ as an additional input. In implementation, the backbone network $F_\theta(\widetilde{\boldsymbol{A}}, \hat{\boldsymbol{A}}, \sigma)$ needs to be changed accordingly and includes a dedicated initial layer to concatenate $\widetilde{\boldsymbol{A}}$ and $\hat{\boldsymbol{A}}$.

During training, however, we do not conduct iterative denoising sampling, and we do not have direct access to $\hat{\boldsymbol{A}}$. Following the heuristics in Chen et al. (2023), we manage $\hat{\boldsymbol{A}}$ during training as follows: given a noisy sample $\widetilde{\boldsymbol{A}}$, with a 50% probability, we set $\hat{\boldsymbol{A}}$ to $\boldsymbol{0}$ (placeholder self-conditioning signal); otherwise, we first obtain $\hat{\boldsymbol{A}}$ by running $D_\theta(\widetilde{\boldsymbol{A}}, \boldsymbol{0}, \sigma)$ to get the denoised result and then use it for self-conditioning. Note that we only add additional information created by the diffusion model itself to the network input, and the overall training loss remains unchanged. During training, when trying to obtain $\hat{\boldsymbol{A}}$, we disable gradient calculation (*e.g.*, using `torch.no_grad()`). Therefore, the extra memory overhead is negligible for training, and, on average, half of the training steps invoke another one-step inference.

---

**Algorithm 1** Sampler w. 2nd-order correction.

---

**Require:** $D_\theta, N, \{t_i\}_{i=0}^{N}, \{\gamma_i\}_{i=0}^{N-1}$.

1: **sample** $\widetilde{\boldsymbol{A}}^{(0)} \sim \mathcal{N}(\boldsymbol{0}, t_0^2 \boldsymbol{I})$, $\widehat{\boldsymbol{A}}_{\text{sc}}^{(0)} = \boldsymbol{0}$.
2: **for** $i = 0$ to $N - 1$ **do**
3:    **sample** $\boldsymbol{\epsilon} \sim \mathcal{N}(\boldsymbol{0}, S_{\text{noise}}^2 \boldsymbol{I})$           ▷ Variance perturbation
4:    $\hat{t}_i \leftarrow (1 + \gamma_i) t_i$               ▷ Time perturbation
5:    $\widetilde{\boldsymbol{A}}^{(\hat{i})} \leftarrow \widetilde{\boldsymbol{A}}^{(i)} + \sqrt{\hat{t}_i^2 - t_i^2}\,\boldsymbol{\epsilon}$       ▷ Noise injection from $t_i$ to $\hat{t}_i$
6:    $\widehat{\boldsymbol{A}}_{\text{sc}}^{(\hat{i})} \leftarrow D_\theta(\widetilde{\boldsymbol{A}}^{(\hat{i})}, \widehat{\boldsymbol{A}}_{\text{sc}}^{(i)}, \hat{t}_i)$     ▷ Self-conditioning signal at current step $i$
7:    $\boldsymbol{d}_i \leftarrow (\widetilde{\boldsymbol{A}}^{(\hat{i})} - \widehat{\boldsymbol{A}}_{\text{sc}}^{(\hat{i})})/\hat{t}_i$        ▷ Evaluate $d\boldsymbol{A}/dt$ at $\hat{t}_i$
8:    $\widetilde{\boldsymbol{A}}^{(i+1)} \leftarrow \widetilde{\boldsymbol{A}}^{(\hat{i})} + (t_{i+1} - \hat{t}_i)\boldsymbol{d}_i$     ▷ Reverse process from $\widetilde{\boldsymbol{A}}^i$ to $\widetilde{\boldsymbol{A}}^{(i+1)}$
   `# Below is 2nd-order correction.`
9:    $\widehat{\boldsymbol{A}}_{\text{sc}}^{(i+1)} \leftarrow D_\theta(\widetilde{\boldsymbol{A}}^{(i+1)}, \widehat{\boldsymbol{A}}_{\text{sc}}^{(\hat{i})}, t_{i+1})$    ▷ Self-conditioning for next step $i + 1$
10:    $\boldsymbol{d}_i' \leftarrow (\widetilde{\boldsymbol{A}}^{(i+1)} - \widehat{\boldsymbol{A}}_{\text{sc}}^{(i+1)})/t_{i+1}$       ▷ Evaluate $d\boldsymbol{A}/dt$ at $t_{i+1}$
11:    $\widetilde{\boldsymbol{A}}^{(i+1)} \leftarrow \widetilde{\boldsymbol{A}}^{(i)} + \frac{1}{2}(t_{i+1} - \hat{t}_i)(\boldsymbol{d}_i + \boldsymbol{d}_i')$   ▷ Heun's integration for $\widetilde{\boldsymbol{A}}^{(i+1)}$ correction
12: **end for**
13: **return** $\widetilde{\boldsymbol{A}}^{(N)}$

---

**Stochastic Sampler with 2nd-order Correction.** Our sampler is formally presented in Alg. 1. It is based on the 2nd-order sampler in Karras et al. (2022). Due to the nature of SDE, one can flexibly select any $N$ sampling steps in the reverse process and their associated SDE time $\{t_i\}_{i=0}^{N}$, which are hyper-parameters of the sampler. We initialize the denoising samples $\widetilde{\boldsymbol{A}}^{(0)}$ from standard normal Gaussians and initialize the self-conditioning signal $\widehat{\boldsymbol{A}}_{\text{sc}}^{(0)}$ as zero tensors. In the reverse process, we always apply a perturbed variance that is slightly larger than the unit variance (L3) to inject Gaussian noise. At the $i$-th iteration, the sampler adds noise to $\widetilde{\boldsymbol{A}}^{(i)}$, moving slightly forward along time to $\hat{t}_i$, whose distance from nominal time $t_i$ is governed by another hyper-parameter $\gamma_i$ (L4-5). We then obtain the self-conditioning signal at L6 for the current perturbed time $\hat{t}_i$ based on the denoised signal at the nominal time $t_i$, which is obtained or initialized prior to the current sampling step. Then, we evaluate $d\boldsymbol{A}/dt$ at perturbed time $\hat{t}_i$, and move the denoising sample $\widetilde{\boldsymbol{A}}^{(\hat{i})}$ backward in time following the reverse process in Eq. (7) to obtain $\widetilde{\boldsymbol{A}}^{(i+1)}$ (L6-8). The sampler then evaluates $d\boldsymbol{A}/dt$ at time $t_{i+1}$ and corrects $\widetilde{\boldsymbol{A}}^{(i+1)}$ using Heun's integration (Süli & Mayers, 2003) on L9-11. Note that we keep track of the generated samples $\widehat{\boldsymbol{A}}_{\text{sc}}^{(i)}$ for model self-conditioning (L9) to save computation.

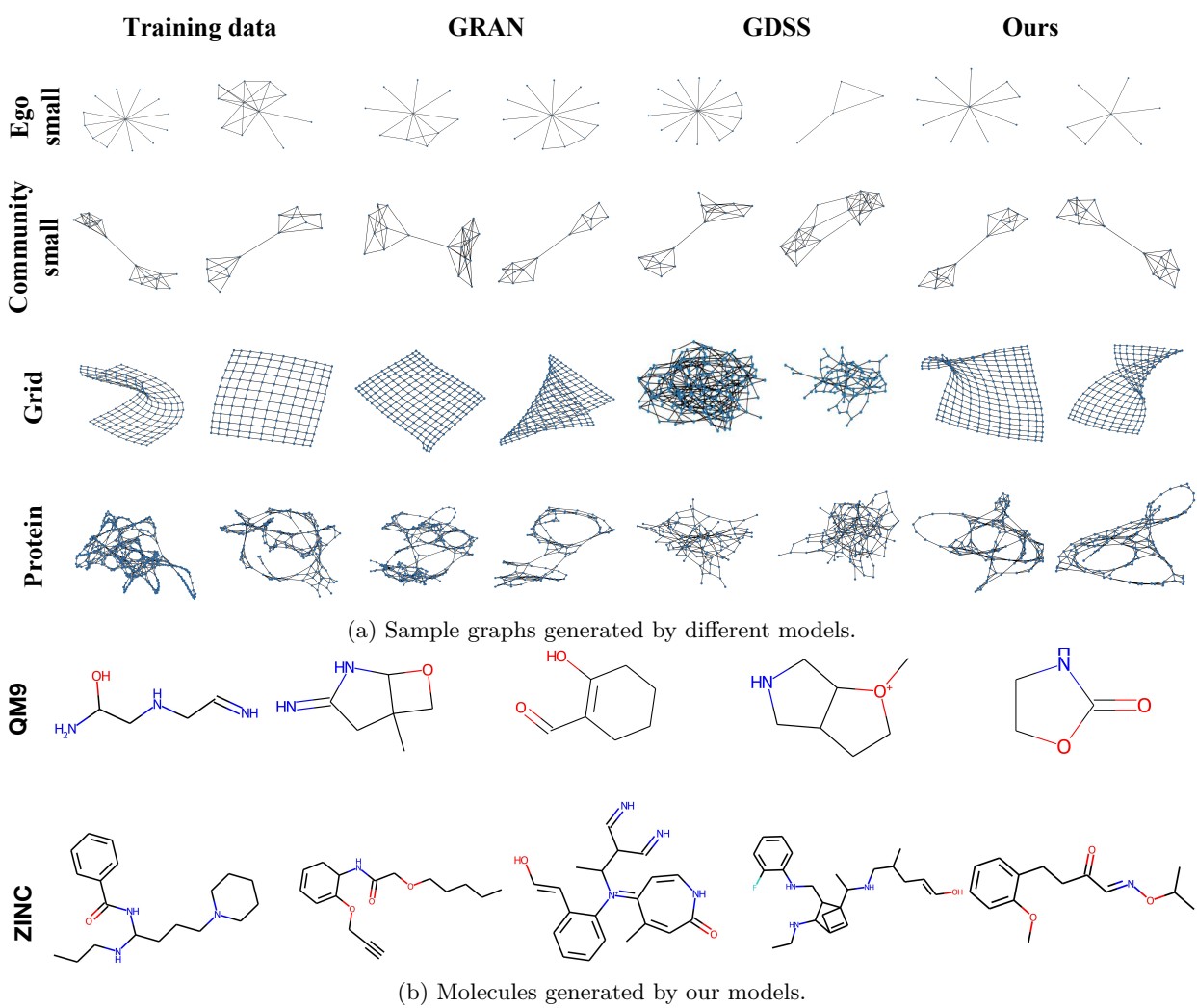

(a) Sample graphs generated by different models.

(b) Molecules generated by our models.

Figure 4: Qualitative results on plain graph and molecule datasets.

# 6 EXPERIMENTS

We now empirically verify the effectiveness of our model on synthetic and real-world graph datasets including molecule generation with node and edge attributes.

**Experiment Setup.** We consider the following synthetic and real-world graph datasets: (1) Ego-small: 200 small ego graphs from Citeseer dataset (Sen et al., 2008), (2) Community-small: 100 random graphs generated by Erdős–Rényi model (Erdős & Rényi, 1959) consisting of two equal-sized communities, (3) Grid: 100 random 2D grid graphs with $|\mathcal{V}| \in [100, 400]$, (4) DD protein dataset (Dobson & Doig, 2003), (5) QM9 dataset (Ramakrishnan et al., 2014), (6) ZINC250k dataset (Irwin et al., 2012).

For synthetic and real-world datasets (1-4), we follow the same setup in Liao et al. (2019); You et al. (2018) and apply random split to use 80% of the graphs for training and the rest 20% for testing. In evaluation, we generate the same number of graphs as the test set to compute the maximum mean discrepancy (MMD) of statistics like node degrees, clustering coefficients, and orbit counts. To compute MMD efficiently, we follow (Liao et al., 2019) and use the total variation distance kernel.

For molecule datasets (5-6), to ensure a fair comparison, we use the same pre-processing and training/testing set splitting as in Jo et al. (2022); Shi* et al. (2020); Luo et al. (2021). We generate 10,000 molecule graphs and compare the following key metrics: (1) validity w/o correction: the proportion of valid molecules without

Table 1: Quantitative results on **ego-small** and **community-small** benchmark datasets.

| Methods | Invariant Loss | Ego-Small ($\mathcal{V} \in [4, 18]$, 200 graphs) | | | Community-Small ($\mathcal{V} \in [12, 20]$, 100 graphs) | | |
| --- | --- | --- | --- | --- | --- | --- | --- |
| | | Deg. ↓ | Clus. ↓ | Orbit. ↓ | Deg. ↓ | Clus. ↓ | Orbit. ↓ |
| GRAN | ✗ | 3.40e-3 | 2.83e-2 | 1.16e-2 | 4.00e-3 | 1.44e-1 | 4.10e-2 |
| EDP-GNN | ✓ | 1.12e-2 | 2.73e-2 | 1.12e-2 | 1.77e-2 | 7.23e-2 | 6.90e-3 |
| GDSS | ✓ | 2.59e-2 | 8.91e-2 | 1.38e-2 | 3.73e-2 | 7.02e-2 | 6.60e-3 |
| DiGress | ✓ | 1.20e-1 | 1.86e-1 | 3.48e-2 | 8.99e-2 | 1.92e-1 | 7.48e-1 |
| GraphVAE-MM | ✗ | 3.45e-2 | 7.32e-1 | 6.39e-1 | 5.87e-2 | 3.56e-1 | 5.32e-2 |
| SPECTRE | ✗ | 4.61e-2 | 1.30e-1 | 6.58e-3 | 8.65e-3 | 7.06e-2 | 8.18e-3 |
| PPGN | ✓ | 2.37e-3 | 3.77e-2 | 4.71e-3 | 1.14e-2 | 1.74e-1 | 4.15e-2 |
| PPGN+PE | ✗ | 1.21e-3 | 3.76e-2 | 4.94e-3 | 2.20e-3 | 9.37e-2 | 3.76e-2 |
| SwinTF | ✗ | 8.50e-3 | 4.42e-2 | 8.00e-3 | 2.70e-3 | 7.11e-2 | **1.30e-3** |
| UNet | ✗ | 9.38e-4 | 2.79e-2 | 6.08e-3 | 4.29e-3 | 7.24e-2 | 8.33e-3 |
| SwinGNN | ✗ | **3.61e-4** | **2.12e-2** | **3.58e-3** | 2.98e-3 | 5.11e-2 | 4.33e-3 |
| SwinGNN-L | ✗ | 5.72e-3 | 3.20e-2 | 5.35e-3 | **1.42e-3** | **4.52e-2** | 6.30e-3 |

valency correction or edge resampling; (2) uniqueness: the proportion of unique and valid molecules; (3) Fréchet ChemNet Distance (FCD) (Preuer et al., 2018): activation difference using pretrained ChemNet; (4) neighborhood subgraph pairwise distance kernel (NSPDK) MMD (Costa & Grave, 2010): graph kernel distance considering subgraph structures and node features.

**Baselines.** We compare our method with state-of-the-art graph generative models. For non-molecule experiments, we include autoregressive model GRAN (Liao et al., 2019), VAE-base GraphVAE-MM (Zahirnia et al., 2022) and GAN-based SPECTRE (Martinkus et al., 2022) as baselines. We also compare with continuous diffusion models with permutation equivariant backbone, *i.e.*, EDP-GNN (Niu et al., 2020) and GDSS (Jo et al., 2022), and discrete diffusion model DiGress (Vignac et al., 2023). We additionally compare to the PPGN networks used in Sec. 4.1. Moreover, to validate the effectiveness of our SwinGNN, we also compare it with a recent UNet (Dhariwal & Nichol, 2021) and the vision SwinTransformer(SwinTF) (Liu et al., 2021) backbones. Specifically, we employ the UperNet on top of the SwinTransformer for the denoising task (refer to App. B.6 for more details). On molecule datasets, we compare with GDSS (Jo et al., 2022), DiGress (Vignac et al., 2023), GraphAF (Shi* et al., 2020) and GraphDF (Luo et al., 2021). We re-run all these baselines with the same data split for a fair comparison.

**Implementation Details.** We try two model variants: standard SwinGNN (60-dim token) and SwinGNN-L (96-dim token), trained using exponential moving average (EMA). The same training and sampling methods are applied to UNet and SwinTF baselines as described in Sec. 5.2. Different node and edge attribute encoding methods, such as scalar, binary bits, and one-hot encoding, are compared in molecule generation experiments. Further details are provided in App. B.

## 6.1 Synthetic Datasets

Fig. 4a showcases samples generated by various models on ego-small, community-small, and grid datasets. The quality of our generated samples is high and comparable to that of auto-regressive models. Invariant diffusion models like GDSS can capture structural information for small graphs but fail on larger and more complicated graphs (*e.g.*, grid). Quantitative maximum mean discrepancy (MMD) results on various graph statistics are presented in Tab. 1 and Tab. 2. Our SwinGNN consistently outperforms the baselines across all datasets by several orders of magnitude, and can generate large graphs with around 400 nodes (grid). Our customized SwinGNN backbone notably outperforms UNet and SwinTF in most metrics, setting itself apart as a more effective solution compared to the powerful visual learning model.

Table 2: Quantitative results on **grid**, and **protein** benchmark datasets.

| Methods | Invariant Loss | Grid ($|\mathcal{V}| \in [100, 400]$, 100 graphs) | | | Protein ($|\mathcal{V}| \in [100, 500]$, 918 graphs) | | |
|---|---|---|---|---|---|---|---|
| | | Deg. ↓ | Clus. ↓ | Orbit. ↓ | Deg. ↓ | Clus. ↓ | Orbit. ↓ |
| GRAN | ✗ | 8.23e-4 | 3.79e-3 | 1.59e-3 | 6.40e-3 | 6.03e-2 | 2.02e-1 |
| EDP-GNN | ✓ | 9.15e-1 | 3.78e-2 | 1.15 | 9.50e-1 | 1.63 | 8.37e-1 |
| GDSS | ✓ | 3.78e-1 | 1.01e-2 | 4.42e-1 | 1.11 | 1.70 | 0.27 |
| DiGress | ✓ | 9.57e-1 | 2.66e-2 | 1.03 | 8.31e-2 | 2.60e-1 | 1.17e-1 |
| GraphVAE-MM | ✗ | 5.90e-4 | **0.00** | 1.60e-3 | 7.95e-3 | 6.33e-2 | 9.24e-2 |
| SPECTRE | ✗ | OOM | OOM | OOM | OOM | OOM | OOM |
| PPGN | ✓ | 1.77e-1 | 1.47e-3 | 1.52e-1 | OOM | OOM | OOM |
| PPGN+PE | ✗ | 8.48e-2 | 5.37e-2 | 7.94e-3 | OOM | OOM | OOM |
| SwinTF | ✗ | 2.50e-3 | 8.78e-5 | 1.25e-2 | 4.99e-2 | 1.32e-1 | 1.56e-1 |
| UNet | ✗ | 9.35e-6 | **0.00** | 6.91e-5 | 5.48e-2 | 7.26e-2 | 2.71e-1 |
| SwinGNN | ✗ | **1.91e-7** | **0.00** | 6.88e-6 | 1.88e-3 | **1.55e-2** | 2.54e-3 |
| SwinGNN-L | ✗ | 2.09e-6 | **0.00** | **9.70e-7** | **1.19e-3** | 1.57e-2 | **8.60e-4** |

Table 3: Quantitative results on **QM9** and **ZINC250k** molecule datasets.

| Methods | Invariant Loss | QM9 ($|\mathcal{V}| \in [1, 9]$, 134,000 molecules) | | | | ZINC250k ($|\mathcal{V}| \in [6, 38]$, 250,000 molecules) | | | |
|---|---|---|---|---|---|---|---|---|---|
| | | Valid w/o cor.↑ | Unique↑ | FCD↓ | NSPDK↓ | Valid w/o cor.↑ | Unique↑ | FCD↓ | NSPDK↓ |
| GraphAF | ✗ | 57.16 | 83.78 | 5.384 | 2.10e-2 | 68.47 | 99.01 | 16.023 | 4.40e-2 |
| GraphDF | ✗ | 79.33 | 95.73 | 11.283 | 7.50e-2 | 41.84 | 93.75 | 40.51 | 3.54e-1 |
| GDSS | ✓ | 90.36 | 94.70 | 2.923 | 4.40e-3 | **97.35** | 99.76 | 11.398 | 1.80e-2 |
| DiGress | ✓ | 95.43 | 93.78 | 0.643 | 7.28e-4 | 84.94 | 99.21 | 4.88 | 8.75e-3 |
| SwinGNN (scalar) | ✗ | 99.68 | 95.92 | 0.169 | 4.02e-4 | 87.74 | **99.98** | 5.219 | 7.52e-3 |
| SwinGNN (bits) | ✗ | 99.91 | 96.29 | 0.142 | 3.44e-4 | 83.50 | 99.97 | 4.536 | 5.61e-3 |
| SwinGNN (one-hot) | ✗ | 99.71 | 96.25 | 0.125 | 3.21e-4 | 81.72 | **99.98** | 5.920 | 6.98e-3 |
| SwinGNN-L (scalar) | ✗ | 99.88 | **96.46** | 0.123 | 2.70e-4 | 93.34 | 99.80 | 2.492 | 3.60e-3 |
| SwinGNN-L (bits) | ✗ | **99.97** | 95.88 | **0.096** | **2.01e-4** | 90.46 | 99.79 | 2.314 | 2.36e-3 |
| SwinGNN-L (one-hot) | ✗ | 99.92 | 96.02 | 0.100 | 2.04e-4 | 90.68 | 99.73 | **1.991** | **1.64e-3** |

## 6.2 Real-world Datasets

**Protein Dataset.** We conduct experiments on the DD protein graph dataset, which has a significantly larger number of nodes (up to 500) and dataset size compared to other benchmark datasets. As shown in Fig. 4a, our model can generate samples visually similar to those from the training set, while previous diffusion models cannot learn the topology well. Tab. 2 demonstrates that the MMD metrics of our model surpass the baselines by several orders of magnitude.

**Molecule Datasets.** Our SwinGNN can be extended to generate molecule graphs with node and edge features (see App. B.3 for details). We evaluate our model against baselines on QM9 and ZINC250k using metrics such as validity without correction, uniqueness, Fréchet ChemNet Distance (FCD) (Preuer et al., 2018), and neighborhood subgraph pairwise distance kernel (NSPDK) MMD (Costa & Grave, 2010). Notably, our models exhibit substantial improvements in FCD and NSPDK metrics, surpassing the baselines by several orders of magnitude. We include the experimental results on novelty scores in App. B.8 for completeness. However, we argue that novelty score may not be a reliable indicator for unconditional generative models, particularly in molecule generation, where novel samples are likely to violate essential chemical principles (Vignac et al., 2023; Vignac & Frossard, 2022).

### 6.3 Ablation Study

### 6.3.1 Model Runtime Efficiency

Table 4: Comparison of running time using one NVIDIA RTX 3090 (24 GB) GPU.

| Method | #params | Training | Inference |
|---|---|---|---|
| GDSS | 0.37M | 215s | 113s |
| DiGress | 18.43M | 713s | 205s |
| Unet | 32.58M | 1573s | 795s |
| SwinGNN | 15.31M | **100s** | **54s** |
| SwinGNN-L | 35.91M | **139s** | **80s** |

In Tab. 4, we compare the training and inference time of various methods also on the grid dataset. With the maximally allowed batch size for each model, we measure the time for training 100 epochs and for generating a fixed number of samples once (equal to the testing set size). Our model operates much more efficiently than baselines handling non-downsampled dense tensors, such as GDSS or DiGress. Specifically, when compared to the DiGress with a similar number of trainable parameters, our model trains approximately 7 times faster and performs inference about 4 times faster on the grid dataset. Our model (SwinGNN-L) sustains a faster training and sampling time, even with the largest parameter size. We also compare the GPU memory use in App. B.5, where our model is more memory-efficient.

### 6.3.2 Diffusion Model Training Techniques

Table 5: Ablations on various training techniques.

| Model | Self-cond. | EMA | Grid ($|\mathcal{V}| \in [100, 400]$, 100 graphs) | | |
|---|---|---|---|---|---|
| | | | Deg. ↓ | Clus. ↓ | Orbit. ↓ |
| GDSS | ✓ | ✓ | 1.82e-1 | 9.70e-3 | 2.28e-1 |
| | ✗ | ✓ | 1.98e-1 | 9.97e-3 | 2.24e-1 |
| | ✓ | ✗ | 3.18e-1 | 1.10e-2 | 3.59e-1 |
| | ✗ | ✗ | 3.78e-1 | 1.01e-2 | 4.42e-1 |
| DiGress | ✓ | ✓ | 3.89e-1 | 1.23e-2 | 0.55 |
| | ✗ | ✓ | 4.92e-1 | 1.02e-2 | 0.59 |
| | ✓ | ✗ | 8.73e-1 | 1.96e-2 | 0.97 |
| | ✗ | ✗ | 9.57e-1 | 2.66e-2 | 1.03 |
| SwinGNN-EDM | ✓ | ✓ | **1.91e-7** | **0.00** | **6.88e-6** |
| | ✗ | ✓ | 1.14e-5 | 2.15e-5 | 1.58e-5 |
| | ✓ | ✗ | 5.12e-5 | 2.43e-5 | 3.38e-5 |
| | ✗ | ✗ | 5.90e-5 | 2.71e-5 | 9.24e-5 |
| SwinGNN-DDPM | ✓ | ✓ | 2.89e-3 | 1.36e-4 | 3.70e-3 |
| | ✗ | ✗ | 4.01e-2 | 8.62e-2 | 1.05e-1 |

In Tab. 5, we perform ablations on grid dataset to assess the impact of self-conditioning and EMA across GDSS, DiGress and our SwinGNN models on the grid dataset. Both tricks slightly improve the baseline models, but they still lag behind our models. Additionally, we compare the EDM (Karras et al., 2022) framework, that incorporates SDE modeling and the objective in Eq. (8), against the vanilla DDPM (Ho et al., 2020), finding that EDM substantially improves performance. Further, we conduct comparison experiments in App. B.6 to showcase our model's superior performance to SwinTF. Discussions on the window size are available in App. B.7.

## 7  CONCLUSION

Invariant graph diffusion models trained with invariant losses face learning challenges from theoretical and empirical aspects. On the other hand, non-invariant models, despite exhibiting strong empirical performance, struggle with invariant sampling. To overcome this, we introduce a simple random permutation technique to help reclaim invariant sampling. We propose a non-invariant model SwinGNN that efficiently approximates 2-WL message passing and can generate large graphs with high qualities. Experiments show that our model achieves state-of-the-art performance in a wide range of datasets. In the future, it is promising to generalize our model to the discrete diffusion framework.

### Acknowledgments and Disclosure of Funding

This work was funded, in part, by NSERC DG Grants (No. RGPIN-2022-04636 and No. RGPIN-2019-05448), the NSERC Collaborative Research and Development Grant (No. CRDPJ 543676-19), the Vector Institute for AI, Canada CIFAR AI Chair, and Oracle Cloud credits. Resources used in preparing this research were provided, in part, by the Province of Ontario, the Government of Canada through the Digital Research Alliance of Canada `alliance.can.ca`, and companies sponsoring the Vector Institute `www.vectorinstitute.ai/#partners`, Advanced Research Computing at the University of British Columbia, and the Oracle for Research program. Additional hardware support was provided by John R. Evans Leaders Fund CFI grant and the Digital Research Alliance of Canada under the Resource Allocation Competition award.

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

**Appendix**

**TABLE OF CONTENTS**

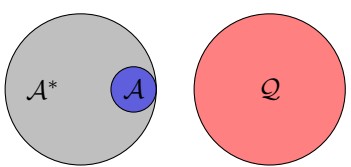 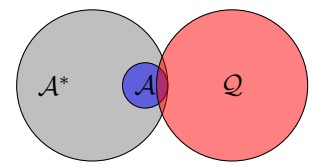 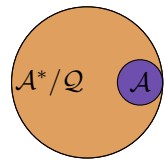

Figure 5: Different configurations of $TV(q, p_{\text{data}})$, given fixed $\mathcal{A}$ (the support of $p_{\text{data}}$) and its induced $\mathcal{A}^*$. Here, we modify $\mathcal{Q}$ (the support of $q$). **Left**: maximal TV, $\mathcal{Q}$ is disjoint from $\mathcal{A}/\mathcal{A}^*$. **Middle**: intermediate TV, intersecting $\mathcal{Q}$ and $\mathcal{A}/\mathcal{A}^*$. **Right**: minimal TV, $\mathcal{Q} = \mathcal{A}^*$.

# A  PROOFS AND ADDITIONAL THEORETICAL ANALYSIS

## A.1  Proof of Lemma 4.1

**Lemma 4.1.** *Assume at least one training graph has $\Omega(n!)$ distinct adjacency matrices in its isomorphism class. Let $\mathcal{P}$ denote all discrete permutation invariant distributions. The closest distributions in $\mathcal{P}$ to $p_{\text{data}}$, measured by total variation, have at least $\Omega(n!)$ modes. If, in addition, we restrict $\mathcal{P}$ to be the set of permutation invariant distributions such that $p(\boldsymbol{A}_i) = p(\boldsymbol{A}_j) > 0$ for all matrices in the training set $\{\boldsymbol{A}_l\}_{l=1}^m$, then the closest distribution is given by $\arg\min_{q \in \mathcal{P}} TV(q, p_{\text{data}}) = p_{\text{data}}^*$.*

*Proof.* Recall the definitions on $p_{\text{data}}$ and $p_{\text{data}}^*$ in our context:

$$p_{\text{data}}(\boldsymbol{A}) = \frac{1}{m} \sum_{i=1}^m \delta(\boldsymbol{A} - \boldsymbol{A}_i), p_{\text{data}}^*(\boldsymbol{A}) = \frac{1}{Z} \sum_{\boldsymbol{A}_i^* \in \mathcal{A}^*} \delta(\boldsymbol{A} - \boldsymbol{A}_i^*)$$

In other words, $p_{\text{data}}$ and $p_{\text{data}}^*$ are both discrete uniform distributions. We use $\mathcal{A}$ and $\mathcal{A}^* = \mathcal{I}_{\boldsymbol{A}_1} \cup \mathcal{I}_{\boldsymbol{A}_2} \cdots \cup \mathcal{I}_{\boldsymbol{A}_m}$ to denote the support of $p_{\text{data}}$ and $p_{\text{data}}^*$ respectively. Let $TV^* := TV(p_{\text{data}}^*, p_{\text{data}})$ be the "golden standard" total variation distance that we aim to outperform.

Let $q \in \mathcal{P}$ without loss of generality. As $q$ is permutation invariant, it must assign equal probability to graphs in the same isomorphism class. We denote the support of $q$ by $\mathcal{Q} = \{\mathcal{I}_{q_i}\}_{i=1}^\Phi$. $q(\boldsymbol{A}) = \sum_{i=1}^\Phi \rho_i \sum_{\boldsymbol{A}_i \in \mathcal{I}_{q_i}} \delta(\boldsymbol{A} - \boldsymbol{A}_i)$, where $\sum_{i=1}^\Phi \rho_i |\mathcal{I}_{q_i}| = 1, \rho_i > 0$, and $\Phi > 0$ stand for the number of isomorphism classes contained in $q$. Fig. 5 summarizes the possibilities of TV used in our proof.

**Proof of $\Omega(n!)$ Modes.** This first part of the lemma imposes no constraints on $\mathcal{P}$: we do not require distributions in $\mathcal{P}$ to be uniform on $\mathcal{A}$. We first prove two helpful claims and then use proof by contradiction to prove our result on $\Omega(n!)$ modes.

Claim 1: The maximal $TV(q, p_{\text{data}})$ is achieved when $\mathcal{Q}$ and $\mathcal{A}^*$ are disjoint (so are $\mathcal{Q}$ and $\mathcal{A}$).

Proof of Claim 1: Without loss of generality, the TV distance is:

$$
\begin{aligned}
TV(q, p_{\text{data}}) &= \sum_{i=1}^\Phi \left( \sum_{\boldsymbol{A} \in \mathcal{I}_{q_i} \cap \mathcal{A}} \left| \rho_i - \frac{1}{|\mathcal{A}|} \right| + \sum_{\boldsymbol{A} \in \mathcal{I}_{q_i}, \boldsymbol{A} \notin \mathcal{A}} \rho_i \right) + \sum_{\boldsymbol{A} \notin \mathcal{Q}, \boldsymbol{A} \in \mathcal{A}} \frac{1}{|\mathcal{A}|} \\
&\leq \sum_{i=1}^\Phi \left( \sum_{\boldsymbol{A} \in \mathcal{I}_{q_i} \cap \mathcal{A}} (\rho_i + \frac{1}{|\mathcal{A}|}) + \sum_{\boldsymbol{A} \in \mathcal{I}_{q_i}, \boldsymbol{A} \notin \mathcal{A}} \rho_i \right) + \sum_{\boldsymbol{A} \notin \mathcal{Q}, \boldsymbol{A} \in \mathcal{A}} \frac{1}{|\mathcal{A}|} \quad \text{(triangle inequality)} \\
&= \sum_{i=1}^\Phi \left( \sum_{\boldsymbol{A} \in \mathcal{I}_{q_i} \cap \mathcal{A}} \frac{1}{|\mathcal{A}|} + \sum_{\boldsymbol{A} \in \mathcal{I}_{q_i}} \rho_i \right) + \sum_{\boldsymbol{A} \notin \mathcal{Q}, \boldsymbol{A} \in \mathcal{A}} \frac{1}{|\mathcal{A}|} \\
&= \sum_{i=1}^\Phi \sum_{\boldsymbol{A} \in \mathcal{I}_{q_i}} \rho_i + \sum_{\boldsymbol{A} \in \mathcal{A}} \frac{1}{|\mathcal{A}|} = 2
\end{aligned}
$$

The triangle inequality is strict when $\sum_{i=1}^\Phi |\mathcal{I}_{q_i} \cap \mathcal{A}| > 0$, as $\rho_i$ and $\frac{1}{|\mathcal{A}|}$ are always positive, and $|\rho_i - \frac{1}{|\mathcal{A}|}| < \rho_i + \frac{1}{|\mathcal{A}|}$ holds. If $\mathcal{Q}$ and $\mathcal{A}^*$ are disjoint, $TV(q, p_{\text{data}}) = \sum_{i=1}^\Phi |\mathcal{I}_{q_i}| \times \rho_i + \frac{1}{|\mathcal{A}|} \times |\mathcal{A}| = 2$, meaning that the maximum TV distance is achieved when there is no intersection between $\mathcal{Q}$ and $\mathcal{A}^*$.

Claim 2: It is always possible to have a $q \in \mathcal{P}$ with $TV(q, p_{\text{data}}) < 2$ by allowing $\mathcal{Q}$ and $\mathcal{A}^*$ to intersect on some isomorphism classes $\{\mathcal{I}_{q_i}\}_{i \in \nu}$. Notice $\mathcal{I}_{q_i} \cap \mathcal{A} \neq \emptyset, \forall i \in \nu$ as per isomorphism.

Proof of Claim 2:

$$
TV(q, p_{\text{data}}) = \sum_{i=1}^{\Phi} \left( \sum_{\boldsymbol{A} \in \mathcal{I}_{q_i} \cap \mathcal{A}} \left| \rho_i - \frac{1}{|\mathcal{A}|} \right| + \sum_{\boldsymbol{A} \in \mathcal{I}_{q_i}, \boldsymbol{A} \notin \mathcal{A}} \rho_i \right) + \sum_{\boldsymbol{A} \notin \mathcal{Q}, \boldsymbol{A} \in \mathcal{A}} \frac{1}{|\mathcal{A}|}
$$

$$
= \sum_{i \in \nu} \left( \sum_{\boldsymbol{A} \in \mathcal{I}_{q_i} \cap \mathcal{A}} \left| \rho_i - \frac{1}{|\mathcal{A}|} \right| + \sum_{\boldsymbol{A} \in \mathcal{I}_{q_i}, \boldsymbol{A} \notin \mathcal{A}} \rho_i \right) + \sum_{i \notin \nu} \sum_{\boldsymbol{A} \in \mathcal{I}_{q_i}} \rho_i + \sum_{\boldsymbol{A} \notin \mathcal{Q}, \boldsymbol{A} \in \mathcal{A}} \frac{1}{|\mathcal{A}|}
$$

$$
= \sum_{i \in \nu} \left( |\mathcal{I}_{q_i} \cap \mathcal{A}| \left| \rho_i - \frac{1}{|\mathcal{A}|} \right| + (|\mathcal{I}_{q_i}| - |\mathcal{I}_{q_i} \cap \mathcal{A}|) \rho_i \right) + \sum_{i \notin \nu} \rho_i |\mathcal{I}_{q_i}| + \sum_{\boldsymbol{A} \notin \mathcal{Q}, \boldsymbol{A} \in \mathcal{A}} \frac{1}{|\mathcal{A}|}
$$

$$
= \sum_{i \in \nu} |\mathcal{I}_{q_i} \cap \mathcal{A}| \left( \left| \rho_i - \frac{1}{|\mathcal{A}|} \right| - \rho_i \right) + \overbrace{\sum_{i=1}^{\Phi} \rho_i |\mathcal{I}_{q_i}|}^{1} + \overbrace{\sum_{\boldsymbol{A} \in \mathcal{A}} \frac{1}{|\mathcal{A}|}}^{1} - \sum_{\boldsymbol{A} \in \mathcal{Q} \cap \mathcal{A}} \frac{1}{|\mathcal{A}|}
$$

$$
= \sum_{i \in \nu} |\mathcal{I}_{q_i} \cap \mathcal{A}| \left( \left| \rho_i - \frac{1}{|\mathcal{A}|} \right| - \rho_i \right) - \sum_{i=1}^{\Phi} |\mathcal{I}_{q_i} \cap \mathcal{A}| \frac{1}{|\mathcal{A}|} + 2
$$

$$
= \sum_{i \in \nu} |\mathcal{I}_{q_i} \cap \mathcal{A}| \overbrace{\left( \left| \rho_i - \frac{1}{|\mathcal{A}|} \right| - \rho_i - \frac{1}{|\mathcal{A}|} \right)}^{<0} - \sum_{i \notin \nu} \overbrace{|\mathcal{I}_{q_i} \cap \mathcal{A}|}^{=0 \text{ if } i \notin \nu} \frac{1}{|\mathcal{A}|} + 2
$$

$$
= \sum_{i \in \nu} |\mathcal{I}_{q_i} \cap \mathcal{A}| \overbrace{\left( \left| \rho_i - \frac{1}{|\mathcal{A}|} \right| - \rho_i - \frac{1}{|\mathcal{A}|} \right)}^{<0} + 2 < 2
$$

The last inequality is strict as $\rho_i$, $\frac{1}{|\mathcal{A}|}$ and $|\mathcal{I}_{q_i} \cap \mathcal{A}|$ are all strictly positive for $i \in \nu$.

Claim 3: Let $q^* \in \arg\min_{q \in \mathcal{P}} TV(q, p_{\text{data}})$ be a minizizer whose support is $\mathcal{Q}^*$. $\mathcal{Q}^*$ and $\mathcal{A}^*$ must not be disjoint, *i.e.*, $\mathcal{Q}^*$ intersects $\mathcal{A}^*$ for at least one isomorphism class in $\mathcal{A}^*$.

Proof of Claim 3: Assume $\mathcal{Q}^*$ and $\mathcal{A}^*$ have no intersection, then $\min_{q \in \mathcal{P}} TV(q, p_{\text{data}}) = TV(q^*, p_{\text{data}}) = 2$, which is validated in Claim 1. From Claim 2, we know there must exist another $q^\dagger \in \mathcal{P}$ with $TV(q^\dagger, p_{\text{data}}) < 2$. Therefore, $\min_{q \in \mathcal{P}} TV(q, p_{\text{data}}) \leq TV(q^\dagger, p_{\text{data}}) < 2$, which contradicts $\min_{q \in \mathcal{P}} TV(q, p_{\text{data}}) = 2$. To minimize $TV(q, p_{\text{data}})$, one can enlarge the set $\nu$ so that the intersection covers $\mathcal{A}$. Consequently, the optimal $|\mathcal{Q}^*|$ has a lower bound $\Omega(n!)$ that does not depend on the size of empirical data distribution $|\mathcal{A}|$.

**Proof of Optimality of $p_{\text{data}}^*$ when $\mathcal{P}$ is Discrete Uniform on a Superset of $\mathcal{A}$.**

Now we proceed to prove the second part of our lemma, which is a special case when $\mathcal{P}$ has some constraints w.r.t. $\mathcal{A}$. Assume $\mathcal{A} = \{\boldsymbol{A}_i\}_{i=1}^m$ consists of $m$ graphs (adjacency matrices) belonging to $k$ isomorphic equivalence classes, where $m \geq k$ due to some potential isomorphic graphs. Subsequently, let $\mathcal{A}^*$ have $l$ adjacency matrices for all $k$ equivalence classes ($l \geq m$), *i.e.*, $\mathcal{A}^* = \cup_{i=1}^m \mathcal{I}_{\boldsymbol{A}_i} = \cup_{i=1}^k \mathcal{I}_{c_i}$, where $\{\mathcal{I}_{c_i}\}_{i=1}^k$ denote $k$ equivalence classes.

Further, let $q_\gamma \in \mathcal{P}$ and let $\mathcal{Q}_\gamma \supseteq \mathcal{A}$ be its support. That is, $q_\gamma = \frac{1}{|\mathcal{Q}_\gamma|} \sum_{\boldsymbol{A}_i \in \mathcal{Q}_\gamma} \delta(\boldsymbol{A} - \boldsymbol{A}_i)$, where $|\mathcal{Q}_\gamma| \geq m$. The total variation distance is:

$$
TV^* = \left| \frac{1}{m} - \frac{1}{l} \right| \times m + \frac{1}{l}(l - m) = 2\left(1 - \frac{m}{l}\right),
$$

$$
TV(q_\gamma, p_{\text{data}}) = \left| \frac{1}{|\mathcal{Q}_\gamma|} - \frac{1}{m} \right| \times m + \left| \frac{|\mathcal{Q}_\gamma| - m}{|\mathcal{Q}_\gamma|} \right| = 2\left(1 - \frac{m}{|\mathcal{Q}_\gamma|}\right).
$$

To minimize $TV(q_\gamma, p_{\text{data}})$ over $q_\gamma$, we need to minimize $|\mathcal{Q}_\gamma|$. Since $\mathcal{A} \subseteq \mathcal{Q}_\gamma$ and $q_\gamma$ is permutation invariant, the smallest $|\mathcal{Q}_\gamma|$ would be $|\cup_{i=1}^m \mathcal{I}_{\boldsymbol{A}_i}| = |\cup_{i=1}^k \mathcal{I}_{c_i}| = |\mathcal{A}^*| = l$. Therefore, we conclude that $\min_{q_\gamma \in \mathcal{P}} TV(q_\gamma, p_{\text{data}}) = TV^* = 2(1 - \frac{m}{l})$, and $\arg\min_{q_\gamma \in \mathcal{P}} TV(q_\gamma, p_{\text{data}}) = p_{\text{data}}^*$.

**Justification on the Constraints of $\mathcal{P}$ to Guarantee the Optimality of $p_{\text{data}}^*$.**

In the end, we justify the reason why $\mathcal{P}$ has to be discrete uniform on a superset of $\mathcal{A}$ (*i.e.*, assign equal probability to each element in $\mathcal{A}$) for the second part of the lemma to hold. We list all possible conditions in the table below and give concrete counterexamples for the cases where the optimality of $p_{\text{data}}^*$ is no longer true, *i.e.*, $p_{\text{data}}^* \notin \arg\min_{q \in \mathcal{P}} TV(q, p_{\text{data}})$ or equivalently $\min_{q \in \mathcal{P}} TV(q, p_{\text{data}}) < TV^* = TV(p_{\text{data}}^*, p_{\text{data}})$. For ease of proof, we further divide $p_{\text{data}}$ into two categories based on the existence of isomorphic graphs.

| | | $p_{\text{data}}$ conditions | |
|---|---|---|---|
| | | $\mathcal{A}$ has isomorphic graphs | $\mathcal{A}$ does not have isomorphic graphs |
| $\mathcal{P}$ conditions | support contains $\mathcal{A}$ and uniform | Our proof in the above: True, $\min_{q \in \mathcal{P}} TV(q, p_{\text{data}}) = TV^*$ | |
| | support contains $\mathcal{A}$ and not uniform | False, Case 1: $\min_{q \in \mathcal{P}} TV(q, p_{\text{data}}) < TV^*$ | False, Case 2: $\min_{q \in \mathcal{P}} TV(q, p_{\text{data}}) < TV^*$ |
| | support does not contain $\mathcal{A}$ | False, Case 3: $\min_{q \in \mathcal{P}} TV(q, p_{\text{data}}) < TV^*$ | False, Case 4: $\min_{q \in \mathcal{P}} TV(q, p_{\text{data}}) < TV^*$ |

Consider graphs with $n = 4$ nodes, let the support of $p_{\text{data}}^*$ (*i.e.*, $\mathcal{A}^*$) be adjacency matrices belonging to two isomorphism classes $\mathcal{I}_a$ and $\mathcal{I}_b$, where $|\mathcal{I}_a| = 24$, $|\mathcal{I}_b| = 6$. Namely, $\mathcal{I}_a$ has no automorphism (*e.g.*, complete disconnected graphs), and the automorphism number of $\mathcal{I}_b$ is 4 (*e.g.*, star graphs). Let $\mathcal{I}_a := \{\boldsymbol{A}_1, \boldsymbol{A}_2, \cdots, \boldsymbol{A}_{24}\}$ and $\mathcal{I}_b := \{\boldsymbol{A}_{25}, \cdots, \boldsymbol{A}_{32}\}$.

Case 1: Let $\mathcal{A} = \{\boldsymbol{A}_{23}, \boldsymbol{A}_{24}, \boldsymbol{A}_{25}\}$ with isomorphic graphs. Let $q_\alpha(\boldsymbol{A}) = \rho_a \sum_{i=1}^{24} \delta(\boldsymbol{A} - \boldsymbol{A}_i) + \rho_b \sum_{i=25}^{32} \delta(\boldsymbol{A} - \boldsymbol{A}_i)$ be a mixture of Dirac delta distributions, where $\rho_a, \rho_b > 0$. Due to normalization, $\sum_{\boldsymbol{A}} q_\alpha(\boldsymbol{A}) = 24\rho_a + 6\rho_b = 1$ or $\rho_b = \frac{1 - 24\rho_a}{6}$. We can tweak $\rho_a, \rho_b$ so that $q_\alpha$ is not necessarily uniform over its support, but $q_\alpha$ is permutation invariant by design, *i.e.*, $q_\alpha(\boldsymbol{A}_1) = q_\alpha(\boldsymbol{A}_2) = \cdots = q_\alpha(\boldsymbol{A}_{24})$ and $q_\alpha(\boldsymbol{A}_{25}) = q_\alpha(\boldsymbol{A}_{26}) = \cdots = q_\alpha(\boldsymbol{A}_{32})$. The TV is:

$$TV^* = |\frac{1}{3} - \frac{1}{32}| \times 3 + \frac{1}{32} \times 29 = \frac{29}{16}, TV(q_\alpha, p_{\text{data}}) = |\rho_a - \frac{1}{3}| \times 2 + |\rho_b - \frac{1}{3}| + 22\rho_a + 5\rho_b = \frac{5}{3} + 4\rho_a.$$

Setting $TV(q_\alpha, p_{\text{data}}) = \frac{5}{3} + 4\rho_a < TV^* = \frac{29}{16}$, we have: $\rho_a < \frac{7}{48}$. Let $\rho_a = \frac{1}{48}, \rho_b = \frac{1}{12}$. We now have: $\min_{q \in \mathcal{P}} TV(q, p_{\text{data}}) \leq TV(q_\alpha, p_{\text{data}}) < TV^*$, and $p_{\text{data}}^*$ is not a minimizer of $\min_{q \in \mathcal{P}} TV(q, p_{\text{data}})$.

Case 2: Let $\mathcal{A} = \{\boldsymbol{A}_{24}, \boldsymbol{A}_{25}\}$ without isomorphic graphs. Similarly, let $q_\alpha(\boldsymbol{A}) = \rho_a \sum_{i=1}^{24} \delta(\boldsymbol{A} - \boldsymbol{A}_i) + \rho_b \sum_{i=25}^{32} \delta(\boldsymbol{A} - \boldsymbol{A}_i)$ be a mixture of Dirac delta distributions. The TV is:

$$TV^* = |\frac{1}{2} - \frac{1}{32}| \times 2 + \frac{1}{32} \times 30 = \frac{15}{8}, TV(q_\alpha, p_{\text{data}}) = |\rho_a - \frac{1}{2}| + |\rho_b - \frac{1}{2}| + 23\rho_a + 5\rho_b = \frac{5}{3} + 6\rho_a.$$

Setting $TV(q_\alpha, p_{\text{data}}) = \frac{5}{3} + 6\rho_a < TV^* = \frac{15}{8}$, we have: $\rho_a < \frac{5}{144}$. Let $\rho_a = \frac{1}{48}, \rho_b = \frac{1}{12}$. Again, we have: $\min_{q \in \mathcal{P}} TV(q, p_{\text{data}}) \leq TV(q_\alpha, p_{\text{data}}) < TV^*$, and $p_{\text{data}}^*$ is not a minimizer of $\min_{q \in \mathcal{P}} TV(q, p_{\text{data}})$.

Case 3: Let $\mathcal{A} = \{\boldsymbol{A}_{23}, \boldsymbol{A}_{24}, \boldsymbol{A}_{25}\}$ with isomorphic graphs. Let $q_\beta$ be a uniform discrete distribution on $\mathcal{I}_b$. $q_\beta$ is permutation invariant (thus $q_\beta \in \mathcal{P}$) whose support does not contain $\mathcal{A}$. The TV is:

$$TV^* = |\frac{1}{3} - \frac{1}{32}| \times 3 + \frac{1}{32} \times 29 = \frac{29}{16}, TV(q_\beta, p_{\text{data}}) = |\frac{1}{6} - \frac{1}{3}| + \frac{1}{6} \times 5 + \frac{1}{3} \times 2 = \frac{5}{3}.$$

So, $\min_{q \in \mathcal{P}^*} TV(q, p_{\text{data}}) \leq TV(q_\beta, p_{\text{data}}) < TV^*$. $p_{\text{data}}^*$ is not a minimizer of $\min_{q \in \mathcal{P}^*} TV(q, p_{\text{data}})$.

Case 4: Let $\mathcal{A} = \{\boldsymbol{A}_{24}, \boldsymbol{A}_{25}\}$ without isomorphic graphs. We use the same $q_\beta$ as above. The TV is:

$$TV^* = |\frac{1}{2} - \frac{1}{32}| \times 2 + \frac{1}{32} \times 30 = \frac{15}{8}, TV(q_\beta, p_{\text{data}}) = |\frac{1}{6} - \frac{1}{2}| + \frac{1}{6} \times 5 + \frac{1}{2} \times 1 = \frac{5}{3}.$$

Again, $\min_{q \in \mathcal{P}^*} TV(q, p_{\text{data}}) \leq TV(q_\beta, p_{\text{data}}) < TV^*$. $p_{\text{data}}^*$ is not a minimizer of $\min_{q \in \mathcal{P}^*} TV(q, p_{\text{data}})$.

In fact, in case 3 and 4, $p_{\text{data}}^* \notin \mathcal{P}$, and by definition, $p_{\text{data}}^*$ cannot be a minimizer of $\min_{q \in \mathcal{P}} TV(q, p_{\text{data}})$. To see that, the support of $p_{\text{data}}^*$ must contain $\mathcal{A}^*$ (a superset of $\mathcal{A}$), while the support of any $q \in \mathcal{P}$ is not a superset of $\mathcal{A}^*$ as per $\mathcal{P}$ conditions in case 3 and 4. $\square$

## A.2 Proof of Lemma 4.2

**Lemma 4.2.** *Let $\boldsymbol{A}$ be a random adjacency matrix distributed according to any graph distribution on $n$ vertices. Let $\boldsymbol{P}_r \sim \mathrm{Unif}(\mathcal{S}_n)$ be uniform over the set of permutation matrices. Then, the induced distribution of the random matrix $\boldsymbol{A}_r = \boldsymbol{P}_r \boldsymbol{A} \boldsymbol{P}_r^\top$, denoted as $q_\theta(\boldsymbol{A}_r)$, is permutation invariant, i.e., $q_\theta(\boldsymbol{A}_r) = q_\theta(\boldsymbol{P}\boldsymbol{A}_r\boldsymbol{P}^\top), \forall \boldsymbol{P} \in \mathcal{S}_n$.*

*Proof.* Let $\mathbb{P}(\cdot)$ denote the probability of a random variable.

$$q_\theta(\boldsymbol{A}_r) = \int q_\theta(\boldsymbol{A}_r|\boldsymbol{A})p_\theta(\boldsymbol{A})d\boldsymbol{A} \qquad \text{(define the random permutation as conditional)}$$

$$= \int \mathbb{P}(P_{\boldsymbol{A}\to\boldsymbol{A}_r})p_\theta(\boldsymbol{A})d\boldsymbol{A} \qquad (P_{\boldsymbol{A}\to\boldsymbol{A}_r} \text{ satifies } P_{\boldsymbol{A}\to\boldsymbol{A}_r}AP^T_{\boldsymbol{A}\to\boldsymbol{A}_r} = \boldsymbol{A}_r)$$

$$= \sum_{\boldsymbol{A}\in\mathcal{I}_{\boldsymbol{A}_r}} \mathbb{P}(P_{\boldsymbol{A}\to\boldsymbol{A}_r})p_\theta(\boldsymbol{A}) \qquad \text{(permutation cannot go beyond isomorphism class)}$$

Let us define the set of 'primitive graphs' that could be generated by $p_\theta$: $\mathcal{C}(\mathcal{I}_{\boldsymbol{A}_r}) = \{\boldsymbol{A}|p_\theta(\boldsymbol{A}) > 0, \boldsymbol{A} \in \mathcal{I}_{\boldsymbol{A}_r}\}$ that corresponds to the isomorphism class $\mathcal{I}_{\boldsymbol{A}_r}$. Let $\mathrm{Aut}(\cdot)$ denote the automorphism number. Then, we have:

$$q_\theta(\boldsymbol{A}_r) = \sum_{\boldsymbol{A}\in\mathcal{I}_{\boldsymbol{A}_r}} \mathbb{P}(P_{\boldsymbol{A}\to\boldsymbol{A}_r})p_\theta(\boldsymbol{A})$$

$$= \sum_{\boldsymbol{A}\in\mathcal{C}(\mathcal{I}_{\boldsymbol{A}_r})} \mathbb{P}(P_{\boldsymbol{A}\to\boldsymbol{A}_r})p_\theta(\boldsymbol{A})$$

$$= \sum_{\boldsymbol{A}_r=\boldsymbol{A},\boldsymbol{A}\in\mathcal{C}(\mathcal{I}_{\boldsymbol{A}_r})} \mathbb{P}(P_{\boldsymbol{A}\to\boldsymbol{A}_r})p_\theta(\boldsymbol{A}) + \sum_{\boldsymbol{A}_r\neq\boldsymbol{A},\boldsymbol{A}\in\mathcal{C}(\mathcal{I}_{\boldsymbol{A}_r})} \mathbb{P}(P_{\boldsymbol{A}\to\boldsymbol{A}_r})p_\theta(\boldsymbol{A})$$

$$= \sum_{\boldsymbol{A}_r=\boldsymbol{A},\boldsymbol{A}\in\mathcal{C}(\mathcal{I}_{\boldsymbol{A}_r})} \frac{\mathrm{Aut}(\boldsymbol{A}_r)}{n!}p_\theta(\boldsymbol{A}) + \sum_{\boldsymbol{A}_r\neq\boldsymbol{A},\boldsymbol{A}\in\mathcal{C}(\mathcal{I}_{\boldsymbol{A}_r})} \frac{|\{\boldsymbol{P}:P_{\boldsymbol{A}\to\boldsymbol{A}_r}AP^T_{\boldsymbol{A}\to\boldsymbol{A}_r} = \boldsymbol{A}_r\}|}{n!}p_\theta(\boldsymbol{A})$$

$$= \sum_{\boldsymbol{A}_r=\boldsymbol{A},\boldsymbol{A}\in\mathcal{C}(\mathcal{I}_{\boldsymbol{A}_r})} \frac{\mathrm{Aut}(\boldsymbol{A}_r)}{n!}p_\theta(\boldsymbol{A}) + \sum_{\boldsymbol{A}_r\neq\boldsymbol{A},\boldsymbol{A}\in\mathcal{C}(\mathcal{I}_{\boldsymbol{A}_r})} \frac{\mathrm{Aut}(\boldsymbol{A}_r)}{n!}p_\theta(\boldsymbol{A})$$

$$= \sum_{\boldsymbol{A}\in\mathcal{C}(\mathcal{I}_{\boldsymbol{A}_r})} \frac{\mathrm{Aut}(\boldsymbol{A}_r)}{n!}p_\theta(\boldsymbol{A})$$

For the case of $\boldsymbol{A}_r = \boldsymbol{A}$, $\mathbb{P}(\boldsymbol{P}_{\boldsymbol{A}\to\boldsymbol{A}_r})$ is the probability of obtaining automorphic permutation matrices, which is $\frac{\mathrm{Aut}(\boldsymbol{A}_r)}{n!}$ by definition. As for $\boldsymbol{A}_r \neq \boldsymbol{A}$, we need to compute the size of $\Omega_{\boldsymbol{P}} = \{\boldsymbol{P} : \boldsymbol{P}_{\boldsymbol{A}\to\boldsymbol{A}_r}AP^T_{\boldsymbol{A}\to\boldsymbol{A}_r} = \boldsymbol{A}_r, \boldsymbol{A}_r \neq \boldsymbol{A}\}$, *i.e.*, how many permutation matrices there are to transform $\boldsymbol{A}$ into $\boldsymbol{A}_r$. The orbit-stabilizer theorem states that there are $\frac{n!}{\mathrm{Aut}(\boldsymbol{A})}$ many distinct adjacency matrices in $\mathcal{I}_{\boldsymbol{A}}$ (size of permutation group orbit). For any $\boldsymbol{A}$, we could divide all the permutation matrices in $\mathcal{S}_n$ into $\rho = \frac{n!}{\mathrm{Aut}(\boldsymbol{A})}$ subgroups, where each group $i \in \{1, 2, \cdots, \rho\}$ transforms $\boldsymbol{A}$ to a new adjacency matrix $\boldsymbol{A}_i$. One of the subgroups transforms $\boldsymbol{A}$ into itself (*i.e.*, automorphism), and the rest $\rho - 1$ subgroups transform $\boldsymbol{A}$ into distinct adjacency matrices. As $\boldsymbol{A}_r \neq \boldsymbol{A}$, the size of $\Omega_{\boldsymbol{P}}$ is equal to the size of one such subgroup, which is $\mathrm{Aut}(\boldsymbol{A})$. So, we could aggregate the two cases of $\boldsymbol{A}_r = \boldsymbol{A}$ and $\boldsymbol{A}_r \neq \boldsymbol{A}$.

For any $\boldsymbol{A}'_r$ and $\boldsymbol{A}_r$ that are isomorphic to each other, we have:

$$q_\theta(\boldsymbol{A}'_r) = \sum_{\boldsymbol{A}\in\mathcal{C}(\mathcal{I}_{\boldsymbol{A}'_r})} \frac{\mathrm{Aut}(\boldsymbol{A}'_r)}{n!}p_\theta(\boldsymbol{A}) = \sum_{\boldsymbol{A}\in\mathcal{C}(\mathcal{I}_{\boldsymbol{A}_r})} \frac{\mathrm{Aut}(\boldsymbol{A}_r)}{n!}p_\theta(\boldsymbol{A}) = q_\theta(\boldsymbol{A}_r)$$

The second equality holds because $\mathrm{Aut}(\boldsymbol{A}'_r) = \mathrm{Aut}(\boldsymbol{A}_r)$, $\mathcal{I}_{\boldsymbol{A}'_r} = \mathcal{I}_{\boldsymbol{A}_r}$, and $\mathcal{C}(\mathcal{I}_{\boldsymbol{A}'_r}) = \mathcal{C}(\mathcal{I}_{\boldsymbol{A}_r})$, which are evident facts for isomorphic graphs. Thus, any two isomorphic graphs have the same probability in $q_\theta$. The random permutation operation propagates the probability of the primitive graphs to all their isomorphic forms. $\square$

### A.3 Invariant Model Distribution via Permutation Equivariant Network

For denoising model, if we consider one noise level, the optimal score network would be the score of the following noisy data distribution $p_\sigma(\tilde{A}) = \frac{1}{m} \sum_{i=1}^m \mathcal{N}(\tilde{A}; A_i, \sigma^2 I)$, which is a GMM with $m$ components for the dataset $\{A_i\}_{i=1}^m$. This is the case for diffusion models with non-permutation-equivariant networks, and in what follows, we first show that for equivariant networks, the noisy data distribution is a GMM with $O(n!m)$ components. As score estimation and denoising diffusion are equivalent, we use the terms 'score' or 'diffusion' interchangeably.

**Lemma A.1.** *Assume we only observe one adjacency matrix out of its isomorphism class in dataset $\{A_i\}_{i=1}^m$, and the size of isomorphism class for each graph is the same, i.e. $|\mathcal{I}_{A_1}| = |\mathcal{I}_{A_2}| = \cdots = |\mathcal{I}_{A_m}|$. Let $s_\theta^{eq}$ be a permutation equivariant score estimator. Under our definitions of $p_{\mathrm{data}}^*(A)$ and $p_{\mathrm{data}}(A)$, the following two training objectives are equivalent:*

$$\mathbb{E}_{p_{\mathrm{data}}^*(A)p_\sigma(\tilde{A}|A)} \left[ \| s_\theta^{eq}(\tilde{A}, \sigma) - \nabla_{\tilde{A}} \log p_\sigma(\tilde{A}|A) \|_F^2 \right] \tag{9}$$

$$= \mathbb{E}_{p_{\mathrm{data}}(A)p_\sigma(\tilde{A}|A)} \left[ \| s_\theta^{eq}(\tilde{A}, \sigma) - \nabla_{\tilde{A}} \log p_\sigma(\tilde{A}|A) \|_F^2 \right]. \tag{10}$$

*Proof.* We conduct the proof from Eq. (10) to Eq. (9). Let $P \in \mathcal{S}_n$ be an arbitrary permutation matrix and $p_{\mathcal{S}_n}$ be a uniform distribution over all possible permutation matrices $\mathcal{S}_n$.

$$\mathbb{E}_{p_{\mathrm{data}}(A)p_\sigma(\tilde{A}|A)}[\| s_\theta^{eq}(\tilde{A}, \sigma) - \nabla_{\tilde{A}} \log p_\sigma(\tilde{A}|A) \|_F^2]$$

$$= \mathbb{E}_{p_{\mathrm{data}}(A)p_\sigma(\tilde{A}|A)}[\| P s_\theta^{eq}(\tilde{A}, \sigma) P^T - P \frac{A - \tilde{A}}{\sigma^2} P^T \|_F^2] \quad \text{( Frobenius norm is permutation invariant)}$$

$$= \mathbb{E}_{p_{\mathrm{data}}(A)p_\sigma(\tilde{A}|A)}[\| s_\theta^{eq}(P\tilde{A}P^T, \sigma) - P \frac{A - \tilde{A}}{\sigma^2} P^T \|_F^2] \quad (s_\theta \text{ is permutation equivariant})$$

$$= \mathbb{E}_{p_{\mathrm{data}}(A)p_\sigma(\tilde{B}|A)}[\| s_\theta^{eq}(\tilde{B}, \sigma) - P \frac{A - P^T \tilde{B} P}{\sigma^2} P^T \|_F^2 \cdot |\mathrm{Det}(\frac{d\tilde{A}}{d\tilde{B}})|] \quad (\text{change of variable } \tilde{B} := P\tilde{A}P^\top)$$

$$= \mathbb{E}_{p_{\mathrm{data}}(A)p_\sigma(\tilde{B}|A)}[\| s_\theta^{eq}(\tilde{B}, \sigma) - \frac{PAP^\top - \tilde{B}}{\sigma^2} \|_F^2 \cdot \overbrace{|\mathrm{Det}(P \otimes P)|}^{=1}]$$

$$= \mathbb{E}_{p_{\mathrm{data}}(A)p_\sigma(\tilde{B}|A)}[\| s_\theta^{eq}(\tilde{B}, \sigma) - \nabla_{\tilde{B}} \log p_\sigma(\tilde{B}|PAP^\top) \|_F^2]$$

$$= \mathbb{E}_{p_{\mathrm{data}}(A)p_{\mathcal{S}_n}(P)p_\sigma(\tilde{B}|A)}[\| s_\theta^{eq}(\tilde{B}, \sigma) - \nabla_{\tilde{B}} \log p_\sigma(\tilde{B}|PAP^\top) \|_F^2] \quad (\text{let } P \sim p_{\mathcal{S}_n} \text{ be uniform})$$

$$= \mathbb{E}_{p_{\mathrm{data}}^*(A)p_\sigma(\tilde{B}|A)}[\| s_\theta^{eq}(\tilde{B}, \sigma) - \nabla_{\tilde{B}} \log p_\sigma(\tilde{B}|A) \|_F^2] \quad (\text{permuting } p_{\mathrm{data}} \text{ samples leads to } p_{\mathrm{data}}^* \text{ samples})$$

$$= \mathbb{E}_{p_{\mathrm{data}}^*(A)p_\sigma(\tilde{A}|A)}[\| s_\theta^{eq}(\tilde{A}, \sigma) - \nabla_{\tilde{A}} \log p_\sigma(\tilde{A}|A) \|_F^2] \quad (\text{change name of random variable})$$

The change of variable between $\tilde{A}$ and $\tilde{B}$ leverages the fact that permuting *i.i.d.* Gaussian random variables does not change the multivariate joint distributions. Conditioned on $A$ and $P$, we have $\tilde{A} = A + \epsilon$; $\tilde{B} = PAP^\top + P\epsilon P^\top$, where the randomness related to $\epsilon$ (Gaussian noise) is not affected by permutation due to *i.i.d.* property. The second last equality due to our definition of $p_{\mathrm{data}}$ and $p_{\mathrm{data}}^*$. Recall we take the Dirac delta function over $\mathcal{A}$ to build $p_{\mathrm{data}}$ and over $\mathcal{A}^*$ to build $p_{\mathrm{data}}^*$, where $\mathcal{A}^*$ is the union of all isomorphism classes in $\mathcal{A}$. By applying random permutation on samples drawn from $p_{\mathrm{data}}$, we subsequently obtain samples following $p_{\mathrm{data}}^*$. The main idea is similar to the proofs in previous works (Niu et al., 2020; Xu et al., 2022; Hoogeboom et al., 2022).

If we do not have the assumptions on the size of isomorphism class, the non-trivial automorphism would make the equality between Eq. (10) and Eq. (9) no longer hold, as each isomorphism class may be weighted differently in the actual invariant distribution. We can then replace the $p_{\mathrm{data}}^*$ by a slightly different invariant distribution: $l$-permuted ($l = n!$) empirical distribution, defined as follows: $p_{\mathrm{data}}^l(A) := \frac{1}{ml} \sum_{i=1}^m \sum_{j=1}^l \delta(A - P_j A_i P_j^\top)$, where $\mathcal{S}_n = \{P_1, \ldots, P_k\}$. Both $p_{\mathrm{data}}^l(A)$ and $p_{\mathrm{data}}^*$ have $O(n!m)$ many modes. Therefore, the assumptions do not affect the number of GMM components for underlying noisy data distribution, which is a result we care about. $\square$

More formally, we connect the number of modes in the discrete distribution $p_{\text{data}}$ or $p_{\text{data}}^*$ to the number of components in their induced GMMs. We show that the noisy data distribution of permutation equivariant network is $p_\sigma^*(\tilde{A}) \coloneqq \frac{1}{Z} \sum_{A_i^* \in \mathcal{A}^*} \mathcal{N}(\tilde{A}; A_i, \sigma^2 I)$ of $O(n!m)$ components. Namely, the optimal solution $s_{\theta^*}^{eq}$ to Eq. (9) or equation 10 is $\nabla_{\tilde{A}} \log p_\sigma^*(\tilde{A})$. Leveraging the results from Vincent (2011), we have

$$\mathbb{E}_{p_{\text{data}}^*(A) p_\sigma(\tilde{A}|A)} \left[ \| s_\theta^{eq}(\tilde{A}, \sigma) - \nabla_{\tilde{A}} \log p_\sigma(\tilde{A}|A) \|_F^2 \right]$$

$$= \overbrace{\mathbb{E}_{p_\sigma^*(\tilde{A})} \left[ \| s_\theta^{eq}(\tilde{A}, \sigma) - \nabla_{\tilde{A}} \log p_\sigma^*(\tilde{A}) \|_F^2 \right]}^{\text{Explicit score matching for } p_\sigma^*(\tilde{A})} - C_1 + C_2,$$

$$C_1 = \mathbb{E}_{p_\sigma^*(\tilde{A})}[\| \nabla_{\tilde{A}} \log p_\sigma(\tilde{A}) \|_F^2], C_2 = \mathbb{E}_{p_{\text{data}}^*(A) p_\sigma(\tilde{A}|A)}[\| \nabla_{\tilde{A}} \log p_\sigma(\tilde{A}|A) \|_F^2].$$

As $C_1, C_2$ are constants irrelevant to $\theta$, optimization objective on $\nabla_{\tilde{A}} \log p_\sigma^*(\tilde{A})$ is equivalent to Eq. (10), the latter of which is often used as the training objective in implementation.

### A.4 Sample Complexity Lower Bound of Non-permutation-equivariant Network

In this part, we study the minimum number of samples (*i.e.*, the sample complexity) required to learn the noisy data distribution in the PAC learning setting. Specifically, our analysis is mainly applicable for the *non-permutation-equivariant* network, where we do not assume any hard-coded permutation symmetry. We leave the analysis for permutation equivariant network as future work.

Recall that training DSM at a single noise level amounts to matching the score of the noisy data distribution. Knowing this sample complexity would help us get a sense of the hardness of training DSM since if you can successfully learn a noisy data distribution then you can obtain its score by taking the gradient. Now we derive a lower bound of the sample complexity for learning the noisy data distribution (*i.e.*, a GMM).

**Lemma A.2.** *Any algorithm that learns the score function of a Gaussian noisy data distribution that contains $l$ centroids of $d$-dimension requires $\Omega(ld/\epsilon_f)$ samples to achieve Fisher information distance $\epsilon_f$ with probability at least $\frac{1}{2}$.*

Here the Fisher divergence (or Fisher information distance) (Johnson & Barron, 2004) is defined as $\mathcal{J}_{\text{F}}(f, g) \coloneqq \mathbb{E}_{f(X)} \left[ \| \nabla_X \log f(X) - \nabla_X \log g(X) \|_F^2 \right]$ where $f(X)$ and $g(X)$ are two absolutely continuous distributions defined over $\mathbb{R}^d$. Now we apply Lemma A.2 to the graph distribution in our context. Recall we assume $m$ graphs $\{\mathcal{A}_i\}_{i=1}^m$ with $n$ nodes in the training set. Similar to our investigation in Sec. 4.1, we use the GMM corresponding to the $l$-permuted empirical distribution: $p_\sigma^l(\tilde{A}) = \frac{1}{ml} \sum_{A \in \cup_{i=1}^m \{P_j A_i P_j^\top\}} \mathcal{N}(\tilde{A}; A, \sigma^2 I)$. Let $q_\theta(A)$ denote the estimated distribution returned by a non-permutation-equivariant network.

**Corollary A.3.** *Any algorithm that learns $q_\theta$ for the target $l$-permuted distribution $p_\sigma^l$ to $\epsilon_f$ error in $\mathcal{J}_F(p_\sigma^l, q_\theta)$ with probability at least $\frac{1}{2}$ requires $\Omega(mln^2/\epsilon_f)$ samples.*

Corollary A.3 states the condition to learn distribution $q_\theta$ explicitly with bounded Fisher divergence, from which one could compute a score estimator $s_{q_\theta} = \nabla_{\tilde{A}} \log q_\theta(\tilde{A})$ with bounded DSM error w.r.t. the target $p_\sigma^l$. The sample complexity lower bound holds regardless of the specific learning algorithm.

The highlight is that the sample complexity lower bound has a dependency on $\Omega(l)$, which would substantially increase as $l$ goes to its maximum $n!$. A more practical implication is that given $\alpha$ training samples drawn from $p_\sigma^l$, one could expect, with at least $\frac{1}{2}$ probability, a score network to have at least $o(\frac{mln^2}{\alpha})$ error in Fisher divergence. If we extend $l$ to the maximum $n!$, the learning bottleneck would be the size of the graph $n$ instead of the number of the graphs $m$, for a single large graph could induce a prohibitively enormous sample complexity. This analysis also aligns with our experimental investigation in Sec. 4.1, where the recall metrics drop with $l$ going up.

**Proof of Lemma A.2.** We first introduce two useful lemmas before diving into the proof.

**Lemma A.4.** *For twice continuously differentiable distributions $f$ and $g$ defined over $\mathbb{R}^d$, let $\mathcal{J}_F(f,g) := \mathbb{E}_{f(\boldsymbol{X})}\left[\|\nabla_{\boldsymbol{X}}\log f(\boldsymbol{X}) - \nabla_{\boldsymbol{X}}\log g(\boldsymbol{X})\|_F^2\right]$ be the Fisher information distance and let $\mathcal{J}_{TV}(f,g) := \sup_{\boldsymbol{B}\subseteq\mathbb{R}^d}\int_{\boldsymbol{B}}(f(\boldsymbol{X}) - g(\boldsymbol{X}))d\boldsymbol{X}$ be the total variation (TV) distance. $\mathcal{J}_F(f,g) \geq C\mathcal{J}_{TV}^2(f,g)$ for some constant $C > 0$. (Huggins et al., 2018; Ley & Swan, 2013). The exact value of $C$ relies on conditions of $f$ and $g$ (*c.f. Theorem 5.3 in Huggins et al. (2018)).*

**Lemma A.5.** *Any method for learning the class of $l$-mixtures of $d$-dimensional isotropic Gaussian distribution with $\epsilon_t$ error in total variation with probability at least $\frac{1}{2}$ has sample complexity $\Omega(ld/\epsilon_t^2)$ (Suresh et al., 2014; Ashtiani et al., 2020).*

Now we restate Lemma A.2 and show the proof formally.

**Lemma A.2.** *Any algorithm that learns the score function of a Gaussian noisy data distribution that contains $l$ centroids of $d$-dimension requires $\Omega(ld/\epsilon_f)$ samples to achieve Fisher information distance $\epsilon_f$ with probability at least $\frac{1}{2}$.*

*Proof.* We derive our results through the distribution learning (*a.k.a.* density estimation) approach on top of existing analysis. Without loss of generality, let $f(\boldsymbol{X})$ and $g(\boldsymbol{X})$ be two twice continuously differentiable distributions defined over $\mathbb{R}^d$. Let us assume $f(\boldsymbol{X})$ to be a GMM whose Gaussian components have isotropic variance, similar to the noisy data distribution $p_\sigma^k$ in the diffusion model. We consider the general distribution learning problem where a learning algorithm takes a sequence of i.i.d. samples drawn from the target distribution $f$ and outputs a distribution $g$ as an estimate for $f$.

According to Lemma A.4, the Fisher information distance $\mathcal{J}_F(f,g)$ is lower bounded by the square of TV distance $\mathcal{J}_{TV}^2(f,g)$ with a positive multiplicative factor $C$, under some mild conditions. In order to bound the $\mathcal{J}_F(f,g)$ by $\epsilon_f$, $\mathcal{J}_{TV}(f,g)$ must be smaller than $\sqrt{\frac{\epsilon_f}{C}}$. Importantly, the target $f$ is a GMM, whose density estimation problem with TV distance have been well-studied and it admits a sample complexity lower bound illustrated in Lemma A.5. Plugging in the desired error bound for $\mathcal{J}_{TV}(f,g) \leq \epsilon_t = \sqrt{\frac{\epsilon_f}{C}}$, we obtain a sample complexity lower bound $\Omega(ld/\epsilon_f)$, where the constant $C$ is absorbed. It has the same PAC-learning meaning for $\mathcal{J}_{TV}(f,g)$ with error bound $\epsilon_t = \sqrt{\frac{\epsilon_f}{C}}$, and for $\mathcal{J}_F(f,g)$ with error bound $\epsilon_f$.

We first identify that TV distance is a weaker metric than the Fisher information distance, the latter of which corresponds to the original score estimation objective. Then, we utilize the recent advances in sample complexity analysis for learning GMMs with bouned TV error, and thus obtain a sample complexity lower bound for score estimation. In summary, we use the result of a 'weaker' distribution learning task to show how hard the score estimation objective at least is. $\square$

**Proof of Corollary A.3.**

**Corollary A.3.** *Any algorithm that learns $q_\theta$ for the target $l$-permuted distribution $p_\sigma^l$ to $\epsilon_f$ error in $\mathcal{J}_F(p_\sigma^l, q_\theta)$ with probability at least $\frac{1}{2}$ requires $\Omega(mln^2/\epsilon_f)$ samples.*

*Proof.* We conduct the proof by applying the results from Lemma A.2. We know $p_\sigma^l$ is an GMM with $O(ml)$ components. Since our samples drawn from the noisy distribution $p_\sigma^l$ are noisy adjacency matrix $\tilde{\boldsymbol{A}} \in \mathbb{R}^{n \times n}$, we first vectorize them to be $\mathbb{R}^{n^2}$. In this way, we can view $p_\sigma^l$ as a GMM with $O(ml)$ components of $n^2$-dimensions. Recall we inject *i.i.d.* noise to each entries, so each Gaussian component has isotropic covariance. The conditions of Lemma A.2 (specifically, Lemma A.5 ) are all satisfied. Plugging in the above parameters, we obtain the sample complexity lower bound $\Omega(mln^2/\epsilon_f)$ for score estimation w.r.t. $p_\sigma^l$ through distribution learning perspective. $\square$

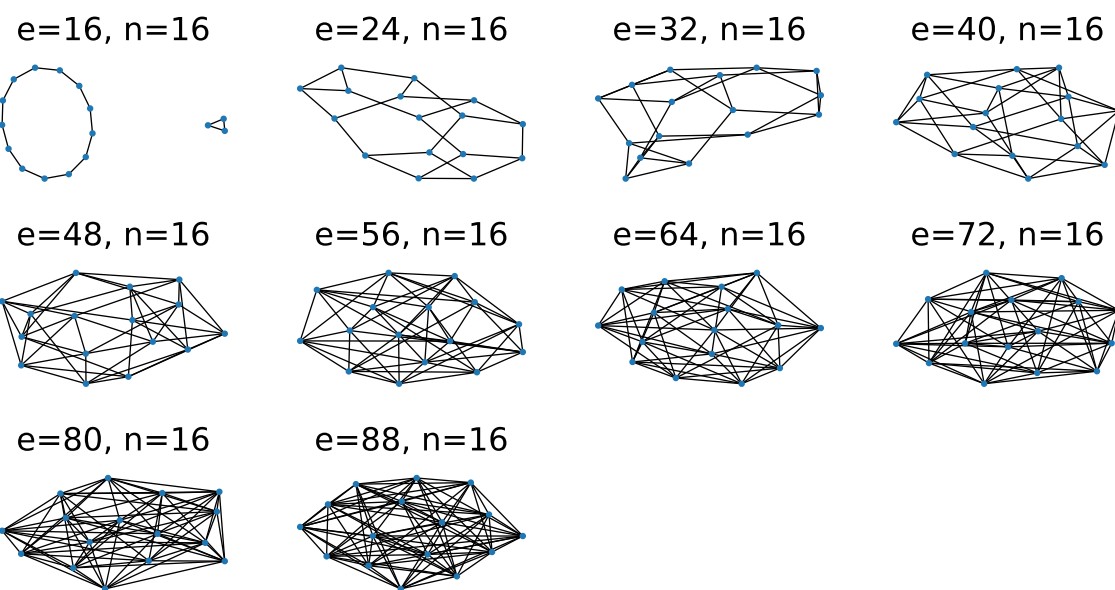

Figure 6: Toy dataset for the investigation of effective target distributions. The dataset comprises 10 randomly generated regular graphs, each with 16 nodes and degrees ranging from 2 to 11.

# B ADDITIONAL EXPERIMENT DETAILS

## B.1 Detailed Experiment Setup

**Toy Dataset.** For the toy dataset experiment, we conduct a sampling process where each model is allowed to generate 100 graphs. To determine the graph recall rate, we perform isomorphism testing utilizing the `networkx`(Hagberg et al., 2008) package. The training set's visualization is provided in Fig. 6.

**Synthetic and Real-world Datasets.** We consider the following synthetic and real-world graph datasets: (1) Ego-small: 200 small ego graphs from Citeseer dataset (Sen et al., 2008) with $|\mathcal{V}| \in [4, 18]$, (2) Community-small: 100 random graphs generated by Erdős–Rényi model (Erdös & Rényi, 1959) consisting of two equal-sized communities whose $|\mathcal{V}| \in [12, 20]$, (3) Grid: 100 random 2D grid graphs with $|\mathcal{V}| \in [100, 400]$. (4) Protein: real-world DD protein dataset (Dobson & Doig, 2003) that has 918 graphs with $|\mathcal{V}| \in [100, 500]$. We follow the same setup in Liao et al. (2019); You et al. (2018) and apply random split to use 80% of the graphs for training and the rest 20% for testing. In evaluation, we generate the same number of graphs as the test set to compute the maximum mean discrepancy (MMD) of statistics like node degrees, clustering coefficients, and orbit counts. To compute MMD efficiently, we follow (Liao et al., 2019) and use the total variation distance kernel.

**Molecule Datasets.** We utilize the QM9 (Ramakrishnan et al., 2014) and ZINC250k (Irwin et al., 2012) as molecule datasets. To ensure a fair comparison, we use the same pre-processing and training/testing set splitting as in Jo et al. (2022); Shi* et al. (2020); Luo et al. (2021). We generate 10,000 molecule graphs and compare the following key metrics: (1) validity w/o correction: the proportion of valid molecules without valency correction or edge resampling; (2) uniqueness: the proportion of unique and valid molecules; (3) Fréchet ChemNet Distance (FCD) (Preuer et al., 2018): activation difference using pretrained ChemNet; (4) neighborhood subgraph pairwise distance kernel (NSPDK) MMD (Costa & Grave, 2010): graph kernel distance considering subgraph structures and node features. We defer the discussion on novelty score in App. B.8.

**Data Quantization.** In this paper, we learn a continuous diffusion model for graph data. Following DDPM (Ho et al., 2020), we map the binary data into the range of [-1, 1] and add noise to the processed data during training. During sampling, we start with Gaussian noise. After the refinement, we map the results from [-1,

1] to [0, 1]. Since graphs are discrete data, we choose 0.5 as a threshold to quantize the continuous results. Similar approaches have been adopted in previous works (Niu et al., 2020; Jo et al., 2022).

## B.2 Network Architecture Details

Table 6: Architecture details of the proposed SwinGNN and major baselines. *The same hyper-parameters are employed for both SwinGNN and SwinGNN-L, barring specific exceptions outlined in the table. The UNet is adopted from Dhariwal & Nichol (2021) with their hyperparameters for the ImageNet-64 dataset.

| | Hyperparameter | Ego-small | Community-small | Grid | DD Protein | QM9 | ZINC250k |
|---|---|---|---|---|---|---|---|
| SwinGNN | Downsampling block layers | [4, 4, 6] | [4, 4, 6] | [1, 1, 3, 1] | [1, 1, 3, 1] | [1, 1, 3, 1] | [1, 1, 3, 1] |
| | Upsampling block layers | [4, 4, 6] | [4, 4, 6] | [1, 1, 3, 1] | [1, 1, 3, 1] | [1, 1, 3, 1] | [1, 1, 3, 1] |
| | Patch size | 3 | 3 | 4 | 4 | 4 | 4 |
| | Window size | 24 | 24 | 6 | 8 | 4 | 5 |
| | Token dimension | 60 | 60 | 60 | 60 | 60 | 60 |
| | Feedforward layer dimension | 240 | 240 | 240 | 240 | 240 | 240 |
| | Number of attention heads | [3, 6, 12, 24] | [3, 6, 12, 24] | [3, 6, 12, 24] | [3, 6, 12, 24] | [3, 6, 12, 24] | [3, 6, 12, 24] |
| | Number of trainable parameters | 15.37M | 15.37M | 15.31M | 15.31M | 15.25M | 15.25M |
| | Number of epochs | 10000 | 10000 | 15000 | 50000 | 5000 | 5000 |
| | EMA | 0.9 | 0.99 | 0.99 | 0.9999 | 0.9999 | 0.9999 |
| SwinGNN-L* | Token dimension | 96 | 96 | 96 | 96 | 96 | 96 |
| | Feedforward layer dimension | 384 | 384 | 384 | 384 | 384 | 384 |
| | Number of trainable parameters | 34.51M | 35.91M | 35.91M | 35.91M | 35.78M | 35.78M |
| | Number of epochs | 10000 | 10000 | 15000 | 50000 | 5000 | 5000 |
| | EMA | 0.99 | 0.95 | 0.95 | 0.9999 | 0.9999 | 0.9999 |
| UNet-ADM* | Channel multiplier | 64 | 64 | 64 | 64 | - | - |
| | Channels per resolution | [1, 2, 3, 4] | [1, 2, 3, 4] | [1, 2, 3, 4] | [1, 1, 1, 1, 2, 4, 6] | - | - |
| | Residule blocks per resolution | 3 | 3 | 3 | 1 | - | - |
| | Number of trainable parameters | 30.86M | 31.12M | 29.50M | 32.58M | - | - |
| | Number of epochs | 5000 | 5000 | 15000 | 10000 | - | - |
| | EMA | 0.9999 | 0.99 | 0.999 | 0.9 | - | - |
| Optimization | Optimizer | Adam | Adam | Adam | Adam | Adam | Adam |
| | Learning rate | $1.0 \times 10^{-4}$ | $1.0 \times 10^{-4}$ | $1.0 \times 10^{-4}$ | $1.0 \times 10^{-4}$ | $1.0 \times 10^{-4}$ | $1.0 \times 10^{-4}$ |

**Our Models and Baselines.** Tab. 6 shows the network architecture details of our models and the UNet baselines on various datasets. Regarding the PPGN (Maron et al., 2019a), we utilize the implementation in Martinkus et al. (2022). It takes an $n \times n$ noisy matrix as input and produces the denoised signal as output. To build non-permutation-equivariant version of PPGN, sinusoidal positional encoding (Vaswani et al., 2017) is applied at each layer. We use the same diffusion setup for PPGN-based networks and our SwinGNN, as specified in Tab. 7. For a fair comparison, we utilize the publicly available code from the other baselines and run experiments using our dataset splits.

**Network Expressivity.** Both theoretical and empirical evidence have underscored the intrinsic connection between the WL test and function approximation capability for GNNs (Mahdavi et al., 2023; Hamilton, 2020; Chen et al., 2019; Morris et al., 2019; Maron et al., 2019b; Xu et al., 2019). The permutation equivariant PPGN layer, notable for its certified 3-WL test capacity, is deemed sufficiently expressive for experimental investigation. Further, it is crucial to note the considerable theoretical expressivity displayed by non-permutation-equivariant GNNs, particularly those with positional encoding (Keriven & Vaiter, 2023; Fereydounian et al., 2022). We argue that the GNNs employed in our studies theoretically have sufficient function approximation capacities, and therefore, the results of our research are not limited by the expressiveness of the network.

## B.3 Node and Edge Attribute Encoding

Molecules possess various edge types, ranging from no bond to single, double, and triple bonds. Also, they encompass diverse node types like C, N, O, F, and others. We employ three methods to encode the diverse node and edge attributes: 1) scalar representation, 2) binary bits, and 3) one-hot encoding.

**Scalar Encoding.** We divide the interval [-1, 1] into several equal-sized sub-intervals (except for the intervals near the boundaries), with each sub-interval representing a specific type. We quantize the node or edge attributes in the samples based on the sub-interval to which it belongs as in Jo et al. (2022).

**Binary-bit Encoding.** Following Chen et al. (2023), we encode attribute integers using multi-channel binary bits. For better training dynamics, we remap the bits from 0/1 to -1/1 representation. During sampling, we perform quantization for the continuous channel-wise bit samples and convert them back to integers.

**One-hot Encoding.** We adopt a similar process as the binary-bit encoding, up until the integer-vector conversion. We use `argmax` to quantize the samples and convert them to integers.

**Network Modifications.** We concatenate the features of the source and target node of an edge to the original edge feature, creating multi-channel edge features as the augmented input. At the final readout layer, we use two MLPs to convert the shared edge features for edge and node denoising.

### B.4 Diffusion Process Hyperparameters

Table 7: Training and sampling hyperparameters in the diffusion process.

$$
\begin{array}{ll}
c_s(\sigma) = \frac{\sigma_d^2}{\sigma_d^2 + \sigma^2} & c_o(\sigma) = \frac{\sigma\sigma_d}{\sqrt{\sigma_d^2 + \sigma^2}} \\
c_i(\sigma) = \frac{1}{\sqrt{\sigma_d^2 + \sigma^2}} & c_n(\sigma) = \frac{1}{4}\ln(\sigma)
\end{array}
$$
$$
\sigma_d = 0.5, \ln(\sigma) \sim \mathcal{N}(P_{\text{mean}}, P_{\text{std}}^2), P_{\text{mean}} = -P_{\text{std}} = -1.2
$$
$$
\sigma_{\text{min}} = 0.002, \sigma_{\text{max}} = 80, \rho = 7
$$
$$
S_{\text{tmin}} = 0.05, S_{\text{tmax}} = 50, S_{\text{noise}} = 1.003, S_{\text{churn}} = 40, N = 256
$$
$$
t_i = (\sigma_{\text{max}}^{\frac{1}{\rho}} + \frac{i}{N-1}(\sigma_{\text{min}}^{\frac{1}{\rho}} - \sigma_{\text{max}}^{\frac{1}{\rho}}))^\rho
$$
$$
\gamma_i = \mathbf{1}_{S_{\text{tmin}} \leq t_i \leq S_{\text{tmax}}} \cdot \min(\frac{S_{\text{churn}}}{N}, \sqrt{2} - 1)
$$

The hyperparameters of the diffusion model training and sampling steps are summarized in Tab. 7. For our SwinGNN model, we maintain a consistent setup throughout the paper, unless stated otherwise. This setup is used for various experiments, including the ablation studies where we compare against the vanilla DDPM (Ho et al., 2020) and the toy dataset experiments.

The pivotal role of refining both the training and sampling phases in diffusion models to bolster performance has been emphasized in prior literature (Song et al., 2021a; Nichol & Dhariwal, 2021; Karras et al., 2022). Such findings, validated across a broad spectrum of fields beyond image generation (Shan et al., 2023; Yang et al., 2022), inspired our adoption of the most recent diffusion model framework. For a detailed discussion on the principles of hyperparameter fine-tuning, readers are encouraged to refer to the previously mentioned studies.

### B.5 Model Memory Efficiency

Table 8: Analysis of the GPU memory consumption during the training phase on a single NVIDIA RTX 3090 (24 GB) graphics card, where OOM stands for out-of-memory. Experiments are performed on the protein dataset that contains 918 graphs, each having a node count ranging from 100 to 500.

| Method | #params | BS=1 | BS=2 | BS=4 | BS=8 | BS=16 | BS=32 |
|--------|---------|------|------|------|------|-------|-------|
| GDSS | 0.37M | 3008M | 4790M | 8644M | 15504M | OOM | OOM |
| DiGress | 18.43M | 16344M | 19422M | 22110M | OOM | OOM | OOM |
| EDP-GNN | 0.09M | 7624M | 13050M | 23848M | OOM | OOM | OOM |
| Unet | 32.58M | 6523M | 10557M | 18247M | OOM | OOM | OOM |
| PPGN | 2.96M | OOM | OOM | OOM | OOM | OOM | OOM |
| PPGN-PE | 3.26M | OOM | OOM | OOM | OOM | OOM | OOM |
| SwinGNN | 15.31M | **2905M** | **3563M** | **5127M** | **8175M** | **14325M** | OOM |
| SwinGNN-L | 35.91M | 4057M | 5203M | 7471M | 12113M | 21451M | OOM |

**GPU Memory Usage.** Our model's efficiency in GPU memory usage during training, thanks to window self-attention and hierarchical graph representations learning, allows for faster training compared to models with similar parameter counts. In Tab. 8, we compare the training memory costs for various models with different batch sizes using the real-world protein dataset.

## B.6 Comparing against the SwinTransformer Baseline

Table 9: Comparing our SwinGNN against the vanilla visual SwinTransformer (Liu et al., 2021).

| Methods | Ego-Small | | | Community-Small | | | Grid | | | Protein | | |
|---------|-----------|--|--|-----------------|--|--|------|--|--|---------|--|--|
| | Deg. ↓ | Clus. ↓ | Orbit. ↓ | Deg. ↓ | Clus. ↓ | Orbit. ↓ | Deg. ↓ | Clus. ↓ | Orbit. ↓ | Deg. ↓ | Clus. ↓ | Orbit. ↓ |
| SwinGNN | **3.61e-4** | **2.12e-2** | **3.58e-3** | 2.98e-3 | 5.11e-2 | **4.33e-3** | **1.91e-7** | **0.00** | 6.88e-6 | 1.88e-3 | **1.55e-2** | 2.54e-3 |
| SwinGNN-L | 5.72e-3 | 3.20e-2 | 5.35e-3 | **1.42e-3** | **4.52e-2** | 6.30e-3 | 2.09e-6 | **0.00** | **9.70e-7** | **1.19e-3** | 1.57e-2 | **8.60e-4** |
| SwinTF | 8.50e-3 | 4.42e-2 | 8.00e-3 | 2.70e-3 | 7.11e-2 | 1.30e-3 | 2.50e-3 | 8.78e-5 | 1.25e-2 | 4.99e-2 | 1.32e-1 | 1.56e-1 |

To further demonstrate the effectiveness of our proposed network in handling adjacency matrices for denoising purposes, we include an additional comparison with SwinTransformer (Liu et al., 2021). SwinTransformer is a general-purpose backbone network commonly used in visual tasks such as semantic segmentation, which also involves dense predictions similar to our denoising task.

In our experiments, we modify the SwinTF + UperNet (Xiao et al., 2018) method and adapt it to output denoising signals. Specifically, we conduct experiments on the various graphs datasets, and the results are presented in Tab. 9. The results clearly demonstrate the superior performance of our proposed SwinGNN model compared to simply adapting the visual SwinTransformer for graph generation.

## B.7 Effects of Window Size

Table 10: Effects of window size on SwinGNN models (grid data).

| Model | Window Size | Grid ($|\mathcal{V}| \in [100, 400]$, 100 graphs) | | |
|-------|-------------|---------|---------|---------|
| | | Deg. ↓ | Clus. ↓ | Orbit. ↓ |
| SwinGNN (ours) | 1 | 1.46e-1 | 7.69e-3 | 2.73e-2 |
| | 2 | 3.06e-4 | 1.75e-4 | 1.77e-4 |
| | 4 | 6.60e-5 | 2.15e-5 | 3.11e-4 |
| | 6 (default) | **1.91e-7** | **0.00** | **6.88e-6** |
| | 12 | 1.92e-5 | **0.00** | 1.50e-5 |

**Receptive Field Analysis.** The window size must be large enough regarding the number of downsampling layers and graph sizes to create a sufficient receptive field. Otherwise, the model would perform poorly (e.g., window size of 1 or 2 in the grid dataset). Let $n$ be the size of the graph and $M$ be the window size. After $k$ iterations of shift-window, the receptive field of each edge token is enlarged $k$ times. The effective receptive field of window attention grows from $M \times M$ into $kM \times kM$. With each half-sizing downsampling operator, the self-attention receptive field grows 2 times larger in the subsequent window attention layer. Putting these together, assume each attention layer has $k$ iterations of window shifting, there are $t$ such layers, each followed by a down-sampling operator, the receptive field is $(2kM)^t$. When $k$, $M$ and $t$ are suitably chosen, it is feasible to ensure the receptive field is larger than the graph size $n$, meaning that the message passing between any two edge tokens can be approximated by our architecture.

**Experimental Results.** Tab. 10 summarizes the impact of window size $M$ on empirical performance in the grid benchmark dataset. When $M$ is considerably small relative to the graph size, the performance tends to be subpar. Conversely, when $M$ is sufficiently large—covering the entire graph within the receptive field with an appropriate number of layers—it acts more like a hyper-parameter, requiring tuning to optimize performance.

### B.8 Additional Results on Molecule Datasets

Table 11: QM9 results with novelty metrics.

| Methods | QM9 | | | | |
| --- | --- | --- | --- | --- | --- |
| | Valid w/o cor.↑ | Novelty | Unique↑ | FCD↓ | NSPDK↓ |
| GraphAF | 57.16 | 81.27 | 83.78 | 5.384 | 2.10e-2 |
| GraphDF | 79.33 | **86.47** | 95.73 | 11.283 | 7.50e-2 |
| GDSS | 90.36 | 65.29 | 94.70 | 2.923 | 4.40e-3 |
| DiGress | 95.43 | 27.69 | 93.78 | 0.643 | 7.28e-4 |
| SwinGNN (scalar) | 99.68 | 15.14 | 95.92 | 0.169 | 4.02e-4 |
| SwinGNN (bits) | 99.91 | 13.60 | 96.29 | 0.142 | 3.44e-4 |
| SwinGNN (one-hot) | 99.71 | 17.34 | 96.25 | 0.125 | 3.21e-4 |
| SwinGNN-L (scalar) | 99.88 | 13.62 | **96.46** | 0.123 | 2.70e-4 |
| SwinGNN-L (bits) | **99.97** | 10.72 | 95.88 | **0.096** | **2.01e-4** |
| SwinGNN-L (one-hot) | 99.92 | 11.36 | 96.02 | 0.100 | 2.04e-4 |
| SwinGNN (scalar, novelty-tuning) | 97.05 | 41.01 | 95.60 | 0.544 | 1.95e-3 |
| SwinGNN-L (scalar, novelty-tuning) | 97.24 | 42.31 | 96.39 | 0.380 | 1.19e-3 |

Table 12: ZINC250k results with novelty metrics.

| Methods | ZINC250k | | | | |
| --- | --- | --- | --- | --- | --- |
| | Valid w/o cor.↑ | Novelty | Unique↑ | FCD↓ | NSPDK↓ |
| GraphAF | 68.47 | **100.0** | 99.01 | 16.023 | 4.40e-2 |
| GraphDF | 41.84 | **100.0** | 93.75 | 40.51 | 3.54e-1 |
| GDSS | **97.35** | **100.0** | 99.76 | 11.398 | 1.80e-2 |
| DiGress | 84.94 | **100.0** | 99.21 | 4.88 | 8.75e-3 |
| SwinGNN (scalar) | 87.74 | 99.38 | **99.98** | 5.219 | 7.52e-3 |
| SwinGNN (bits) | 83.50 | 99.29 | 99.97 | 4.536 | 5.61e-3 |
| SwinGNN (one-hot) | 81.72 | 99.91 | **99.98** | 5.920 | 6.98e-3 |
| SwinGNN-L (scalar) | 93.34 | 95.43 | 99.80 | 2.492 | 3.60e-3 |
| SwinGNN-L (bits) | 90.46 | 95.73 | 99.79 | 2.314 | 2.36e-3 |
| SwinGNN-L (one-hot) | 90.68 | 96.39 | 99.73 | **1.991** | **1.64e-3** |
| SwinGNN (scalar, novelty-tuning) | 88.13 | **100.0** | 99.94 | 5.43 | 7.75e-3 |
| SwinGNN-L (scalar, novelty-tuning) | 90.78 | 99.0 | 99.95 | 3.68 | 5.37e-3 |

**Discussion on Novelty Metric.** In the main paper, we do not report novelty on the molecule datasets following Vignac et al. (2023); Vignac & Frossard (2022). Novelty metric is measured by proportion of generated samples not seen in the training set. The QM9 dataset provides a comprehensive collection of small molecules that meet specific predefined criteria. Generating molecules outside this set (*i.e.*, novel graphs) does not necessarily indicate that the network accurately capture the underlying data distribution. Further, in ZINC250k dataset, all models show very high novelty, making novelty not distinguishable for model performance.

Moreover, we argue that training the diffusion models is essentially maximizing the lower bound of likelihood. Diffusion (*i.e.*, score-based) models are essentially maximum likelihood estimators (MLE) (Hyvärinen, 2005). Generating samples resembling the training data is actually consistent with the MLE objective, as theoretically the closest model distribution to the training distribution would be the Dirac delta functions of training data. Novelty metric may not be a good indicator of model performance on its own. For example, a poorly trained generative model may have very high novelty and bad performance.

**Results with Novelty Metric.** Nevertheless, for the sake of completeness, we present additional experimental results including the novelty metrics in the QM9 and ZINC250k molecule datasets, as seen in and Tab. 12. In the main experimental section (Tab. 3), the optimization of training hyperparameters is mainly directed towards improving FCD and NSPDK metrics, with the novelty scores presented in the initial rows. When the novelty scores are considered in the hyperparameter tuning, the results for the scalar-encoding SwinGNN models are displayed in the concluding rows, demonstrating that our models still maintain remarkable performance.

On the QM9 dataset (Tab. 11), our model outperforms the DiGress baseline by exhibiting higher novelty and excelling in FCD, no matter if it is fine-tuned for novelty or not. In comparison, GraphAF, GraphDF, and GDSS models, despite their competitive uniqueness and novelty, demonstrate deficiencies in validity, FCD, or NSPDK scores. Specifically, their high novelty suggests the generation of a substantial number of new, likely undesirable molecules that violate fundamental chemical principles. This is indicative of a significant deviation from the training distribution, as reflected in their low validity and elevated FCD and NSPDK scores.

In the ZINC250k dataset (Tab. 12), all models exhibit a high novelty, scoring at least 95%. However, the baseline models struggle in aspects of validity, FCD, and NSPDK scores, indicating difficulties in effectively capturing the data distribution. Contrarily, our model (SwinGNN-L) stands out by achieving the best FCD and NSPDK scores when not specifically optimized for novelty. Furthermore, with dedicated tuning, our model's novelty score closely competes with others (*e.g.*, 99.0 vs 100.0), demonstrating a harmonious balance between fostering novelty and maintaining crucial chemical attributes in the generated molecules.

### B.9 Additional Qualitative Results

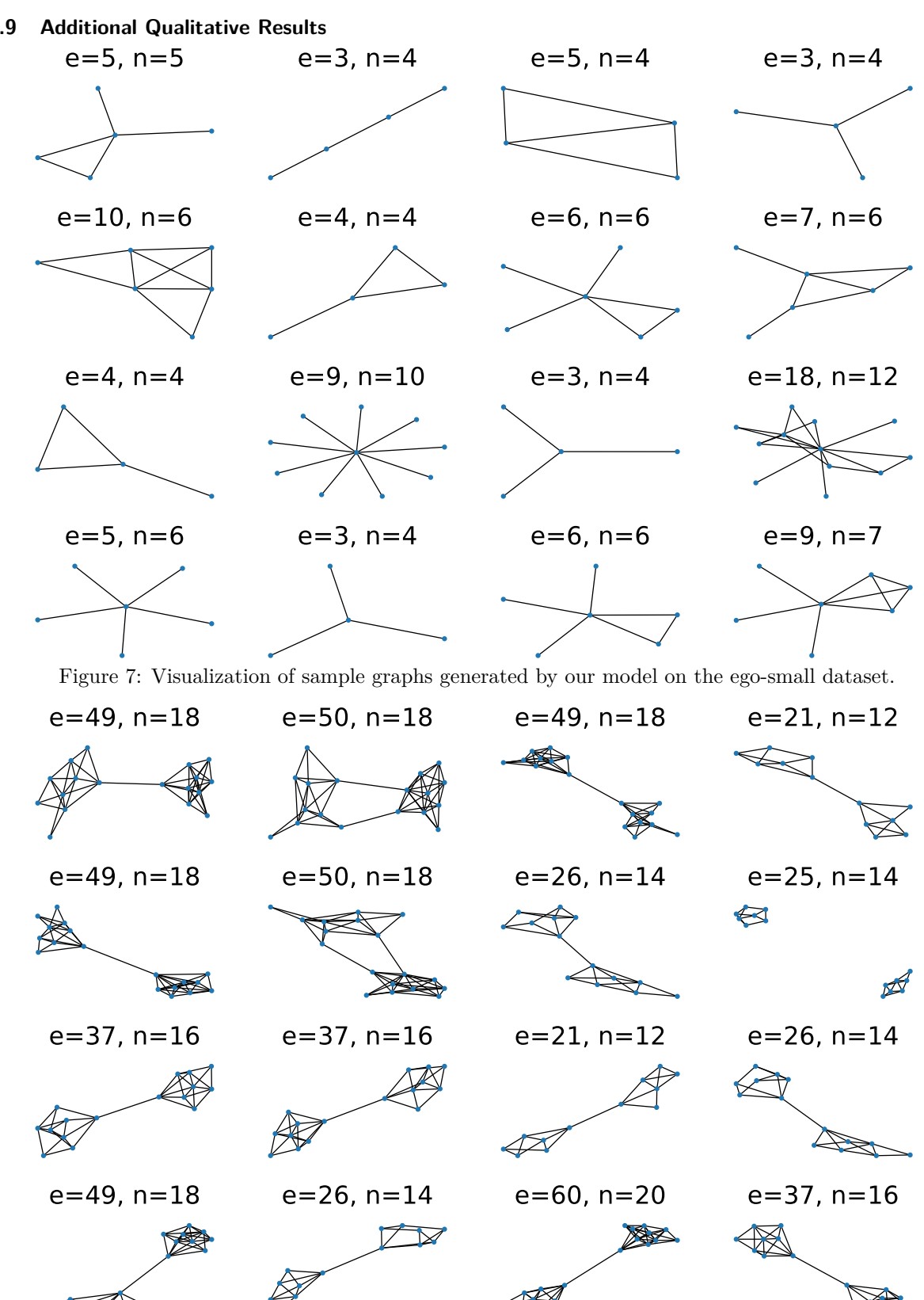

Figure 7: Visualization of sample graphs generated by our model on the ego-small dataset.

Figure 8: Visualization of sample graphs generated by our model on the community-small dataset.

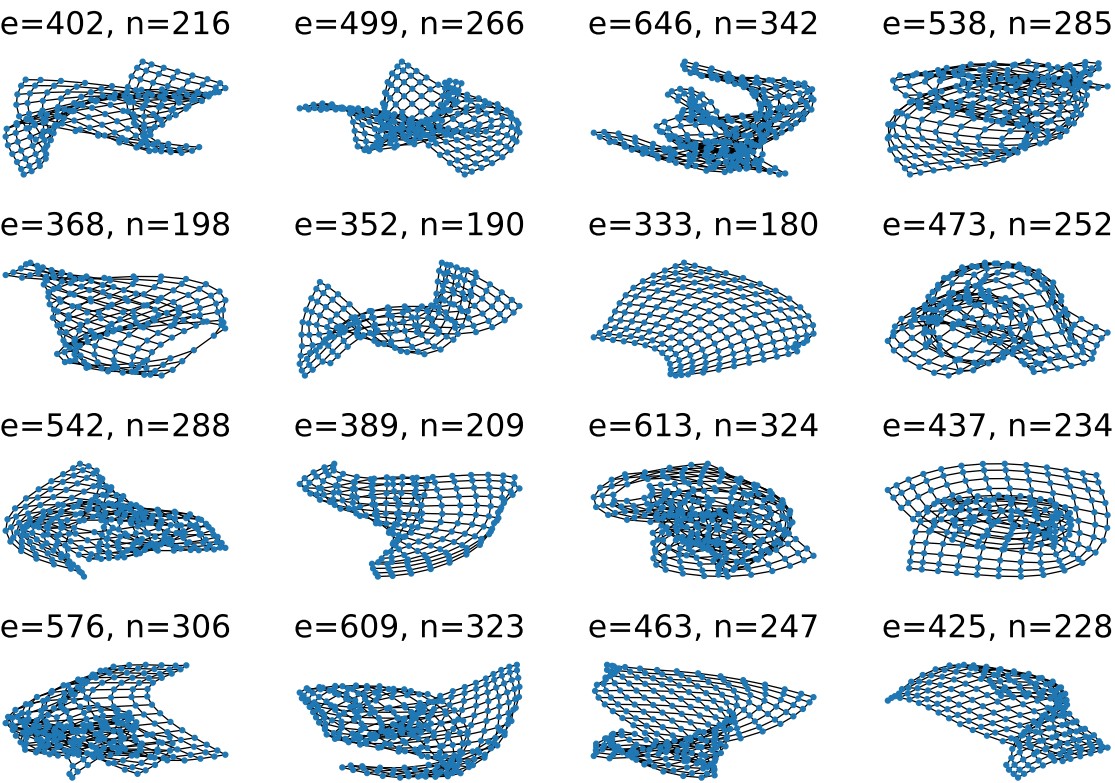

Figure 9: Visualization of sample graphs generated by our model on the grid dataset.

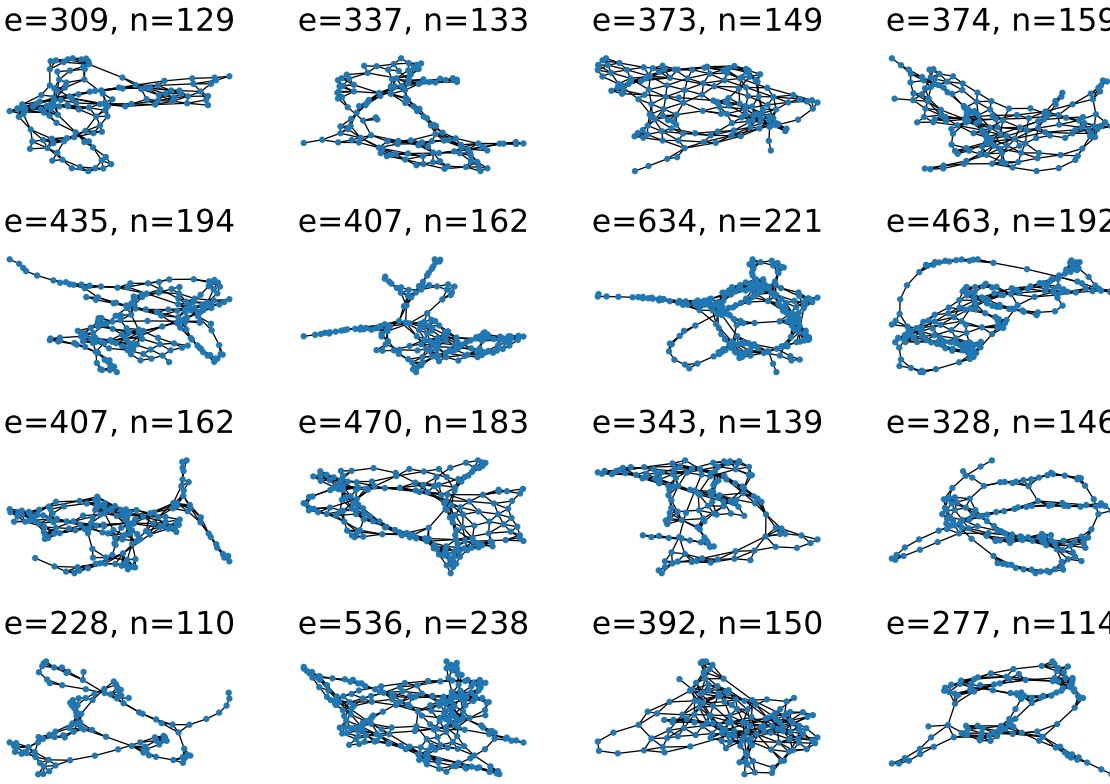

Figure 10: Visualization of sample graphs generated by our model on the DD protein dataset.

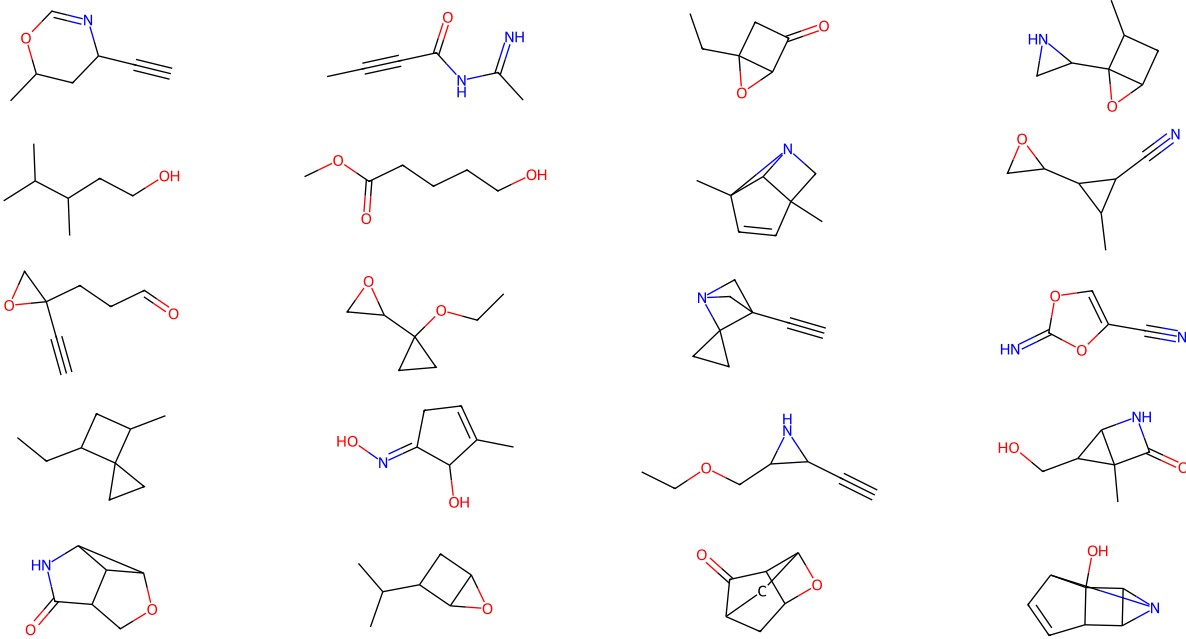

Figure 11: Visualization of sample graphs generated by our model on the QM9 molecule dataset.

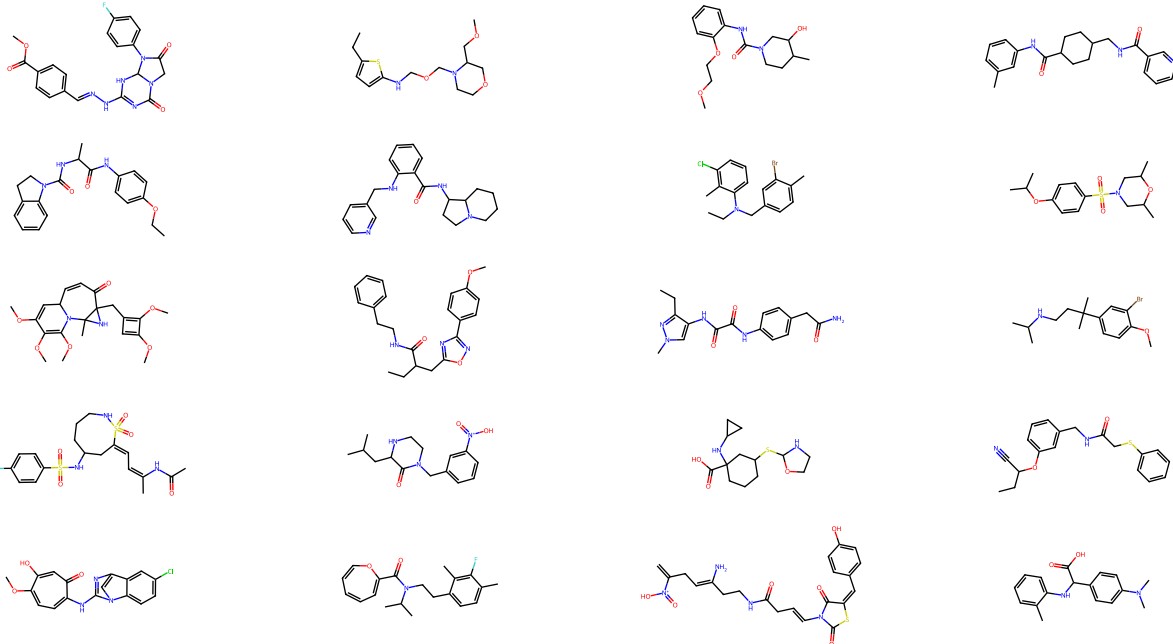

Figure 12: Visualization of sample graphs generated by our model on the ZINC250k molecule dataset.

