# OpenReview forum: "SwinGNN: Rethinking Permutation Invariance in Diffusion Models for Graph Generation"
_TMLR — Accepted by TMLR_

### Review · Reviewer_hsNm · 2024-04-03

**Summary Of Contributions:**

This paper looks at the problem of generating graphs. The paper claims that due to the permutation invariance of the graph, this problem is significantly challenging. In particular, prior work has attempted to tackle this by using a permutation equivariant neural network and a permutation invariant loss function.

However, the paper claims that this is suboptimal. The reason provided is that for most datasets, we only see one instance (not all permutations). Then due to the permutation invariance the model "incorrectly" assigns non-zero probabilities to the permutations. Further, if we try to correct this by adding the permutations to the dataset, we create too many modes and hence make optimization difficult.

The paper then proposes an architecture and tests it for real and synthetic datasets. The paper compares against a variety of different methods and shows and improvement.

**Audience:**

Yes

**Claims And Evidence:**

Yes

**Requested Changes:**

Please see the weaknesses and add more details about the model and the training process.

**Strengths And Weaknesses:**

**Strengths**

The paper tackles an important problem of graph generation and studies the interesting phenomenon of equivariance and invariance in the generation process.

The paper presents a simple and interesting argument for why equip/invariance might be a bad thing.

The proposed method seems to do well.

**Weaknesses**

I think the writing of the paper could be improved. Specifically since TMLR doesn't have a page limit. In particular I think section 5.2 needs a lot of details, specially the two sections on the self conditioning and the stochastic sampler.

The other weaknesses (sort of the related to the first) is it is unclear to me why the network has certain features. Specifically,

1) The locality window are completely arbitrary and do not reflect locality on the graph. The splitting along even and odd party indexes does not help with this. Why is this is a reasonable thing to do?

2) The loss function in Equation 4. The paper mentions that at $t=T$, this is pure Gaussian noise, however, it is still just a picture of Gaussian, just with bigger variance. Shouldn't the means be updated through time as well?

3) The self conditioning and the sampler need more details.

Hence, it is not clear to me what is new in the paper.

**Questions**

If the network were invariant and not equivariant. Would this alleviate the problem?

The arguments for the modes only works for methods that find the density of the model right - such as diffusion models, and flow based models. It does not apply to methods (eg. GANS) that do not do density estimation right?

---

> ### Author Response · Authors · 2024-05-12
>
> > $\textbf{Q6@hsNm}$: I think the writing of the paper could be improved. Specifically since TMLR doesn't have a page limit. In particular I think section 5.2 needs a lot of details, specially the two sections on the self conditioning and the stochastic sampler.
>
> $\textbf{A6@hsNm}$: Thanks for your comment. We have expanded the text in Section 5.2 to clarify the technical details of self-conditioning and the second-order sampler. Please check our latest PDF for updates.
>
> > $\textbf{Q7@hsNm}$: The locality window are completely arbitrary and do not reflect locality on the graph. The splitting along even and odd party indexes does not help with this. Why is this is a reasonable thing to do?
>
> $\textbf{A7@hsNm}$: We acknowledge that both the locality window and the splitting based on index parity are agnostic to the graph structure. This is because, in the diffusion process, there is no clear graph topology in the denoiser network input due to the injected Gaussian noise. For instance, in the reverse process, the adjacency matrix is transformed from pure Gaussian noise to have meaningful topology. The denoiser network must handle the noisy adjacency matrix throughout the sampling process, even before any clean graph topology can be perceived. Therefore, we view noisy input data $\tilde{\boldsymbol{A}} \in \mathbb{R} ^{n \times n}$ as weighted fully-connected graphs with $n$ nodes. Applying the grid-based window attention with shifting mechanism can help improve the message passing for the fully-connected graphs, ensuring interactions between every two nodes are modeled without incurring high computation costs.
>
> > $\textbf{Q8@hsNm}$: The loss function in Equation 4. The paper mentions that at t=T, this is pure Gaussian noise, however, it is still just a picture of Gaussian, just with bigger variance. Shouldn't the means be updated through time as well?
>
> $\textbf{A8@hsNm}$: Thanks for the comment. Indeed, in the DDPM [1] forward process, the means are downscaled and gradually reduced to zero as the diffusion time step increases to $T$. Here, we follow the SDE formulations in EDM [2] to use a slightly different forward process.
>
> We would like to point out that, in implementations, we rescale the range of the adjacency matrix data into [-1, 1].
> With a large enough variance, the $p_{\sigma(T)}$ (an $m$-component GMMs) closely resembles pure Gaussian noise without altering the means.
>
> > $\textbf{Q9@hsNm}$: The self conditioning and the sampler need more details.
>
> $\textbf{A9@hsNm}$: Thank you for your comment. We have expanded the text in Section 5.2 to clarify the technical details of self-conditioning and the second-order sampler.
>
> > $\textbf{Q10@hsNm}$: Hence, it is not clear to me what is new in the paper.
>
> $\textbf{A10@hsNM}$: Our contributions are: 1) identifying the challenges of learning permutation-invariant target distributions; 2) providing insights into graph generative model architecture design (for example, not restricted to equivariant architecture); and 3) proposing a model that achieves state-of-the-art performance on various synthetic and real-world datasets, including plain graphs and molecules.
>
> > $\textbf{Q11@hsNm}$: If the network were invariant and not equivariant. Would this alleviate the problem?
>
> $\textbf{A11@hsNm}$:
> Thank you for raising this interesting question. From a theoretical perspective, we conjecture that invariant networks may perform better than equivariant networks but not better than non-equivariant networks without any architectural constraints.
>
> In our analysis, a permutation-invariant network means that the matrix-to-matrix denoising function is invariant to input node ordering. In other words, permuted noisy inputs result in exactly the same denoised matrices.
>
> First, an invariant network cannot mathematically enforce the learning of permutation-invariant distributions. In contrast to equivariant networks, the effective target distribution of an invariant network has far fewer modes and is less complex (as shown in Lemma 4.1). Therefore, fitting the target distribution could arguably be easier for invariant networks.
>
> However, invariant networks are a subset of non-equivariant neural functions, theoretically limiting their design space. In practice, matrix-to-matrix invariant networks offer far fewer design choices than general non-equivariant networks.
>
> We believe this comment raises an interesting problem with both theoretical and empirical significance, and we will investigate it further in the future.
>
> \
> [1] Ho, J., Jain, A. and Abbeel, P., 2020. Denoising diffusion probabilistic models. Advances in neural information processing systems, 33, pp.6840-6851. \
> [2] Karras, T., Aittala, M., Aila, T. and Laine, S., 2022. Elucidating the design space of diffusion-based generative models. Advances in Neural Information Processing Systems, 35, pp.26565-26577.

---

> > ### Author Response · Authors · 2024-05-12
> >
> > > $\textbf{Q12@hsNm}$: The arguments for the modes only works for methods that find the density of the model right - such as diffusion models, and flow based models. It does not apply to methods (eg. GANS) that do not do density estimation right?
> >
> > $\textbf{A12@hsNm}$: Thank you for the insightful comment. We would like to clarify that our arguments regarding target distribution modes apply to any model that can rigorously learn invariant distributions, regardless of its ability to conduct density estimation.
> >
> > We acknowledge that existing invariant models are predominantly based on diffusion or flow, and they do permit density estimation. However, we do not consider this a necessary assumption in our mode analysis.
> >
> > For generative models, the correlation between permutation invariance and density estimation capability remains an open question. We will investigate this theoretical correlation further in future work.

---

> > ### Comment · Reviewer_hsNm · 2024-06-03
> > **Thanks**
> >
> > Thank you for the clarifying comments!

---

> > > ### Author Response · Authors · 2024-06-11
> > >
> > > Thanks for your response!

---

### Review · Reviewer_Asq3 · 2024-04-04

**Summary Of Contributions:**

The paper first establishes that the degradation in performance for permutation-invariant diffusion models of graphs is not only due to restrictive architecture designs but also because of the increasing modes of target distributions. Then, the authors introduce SwinGNN, a non-invariant diffusion model. SwinGNN incorporates an efficient 2-WL message-passing network using the novel shifted-window-based self-attention mechanism from SwinGNNs, which allows it to scale and generate large graphs effectively. Additionally, the paper demonstrates that random permutation of generated graphs can be a simple yet effective technique to achieve permutation-invariant sampling without necessarily relying on an invariant loss function. The authors have presented many experiments to validate the performance of SwinGNN, which showed that it outperforms existing methods by a significant margin on both synthetic and real-world datasets, including protein and molecule datasets, and it can be conducted more efficiently than other baselines.

**Audience:**

Yes

**Claims And Evidence:**

Yes

**Requested Changes:**

1. Can you explain the notation $\mathcal{N}(\tilde{A};A,\sigma^2I)$?

2. In Figure 2, did you train the models with the same hyperparameters for all different $l$? How do you choose distinct permutation matrices $P_1,\ldots,P_l$ in the experiments?

**Strengths And Weaknesses:**

### Strengths:

1. The research provides both theoretical analysis and empirical evidence. The theory part is clear and well-written. SwinGNN uses a simple random permutation technique that ensures invariant sampling without the need for an invariant loss, providing flexibility in model architecture design.

2. The paper has conducted extensive experiments to compare SwinGNN with other baselines. For instance, this paper highlights that SwinGNN maintains a balance between fostering novelty and preserving important chemical properties in molecule generation, outperforming other models in validity, FCD, and NSPDK scores, especially in the QM9 dataset. SwinGNN is noted to be memory-efficient during training, which is crucial for practical deployment and experimentation on various graph sizes and types.


### Weaknesses:

The presentation in Section 5.1 about the derivation of the architecture of SwinGNN needs to be expanded. There should be a brief introduction of $k$-order GNN and the $k$-WL GNN for readers unfamiliar with these concepts. Explaining these concepts would lay the groundwork for why SwinGNN’s architecture is structured the way it is, and why certain architectural choices were made, such as using shifted window-based self-attention to approximate the 2-WL message passing. Besides, there should be more details about the comparison between SwinGNN and Swin Transformers to show the expressivity of the SwinGNN.

---

> ### Author Response · Authors · 2024-05-12
>
> > $\\textbf{Q3@Asq3}$: The presentation in Section 5.1 about the derivation of the architecture of SwinGNN needs to be expanded. There should be a brief introduction of k-order GNN and the k-WL GNN for readers unfamiliar with these concepts. Explaining these concepts would lay the groundwork for why SwinGNN’s architecture is structured the way it is, and why certain architectural choices were made, such as using shifted window-based self-attention to approximate the 2-WL message passing. Besides, there should be more details about the comparison between SwinGNN and Swin Transformers to show the expressivity of the SwinGNN.
>
> $\\textbf{A3@Asq3}$: Thank you for your comment. As suggested, we have added more explanation in Section 5.1 to introduce the $k$-WL GNNs and highlight comparisons with SwinTransformers. Please check our latest PDF for updates.
>
> > $\\textbf{Q4@Asq3}$: Can you explain the notation $\\mathcal{N}(\\tilde{\\boldsymbol{A}}; \\boldsymbol{A}, \\sigma^2 \\boldsymbol{I})$?
>
> $\\textbf{A4@Asq3}$: Thank you for your comment. The notation $\\mathcal{N}(\\tilde{\\boldsymbol{A}}; \\boldsymbol{A}, \\sigma^2 \\boldsymbol{I})$ means that the random variable $\\tilde{\\boldsymbol{A}}$ follows a Gaussian distribution with a mean of $\\boldsymbol{A}$ and a variance matrix of $\\sigma^2 \\boldsymbol{I}$, where $\\boldsymbol{I}$ indicates that the covariance is isotropic.
>
>
> > $\\textbf{Q5@Asq3}$: In Figure 2, did you train the models with the same hyperparameters for all different $l$? How do you choose distinct permutation matrices $P_1, …, P_l$ in the experiments?
>
> $\\textbf{A5@Asq3}$: We controlled the hyperparameters for a fair comparison. For each method, we used the same hyperparameters for different $\\boldsymbol{I}$. Since the toy dataset is rather small, we found the hyperparameter tuning does not result in a significant performance difference, as long as the training reaches convergence.
>
> For the permutations, we randomly draw $l$ permutation matrices and do rejection sampling to ensure they are unique, i.e., keep drawing permutation matrices and removing the identical ones until we have $l$ distinct permutation matrices.
> To ensure a fair comparison, we only save one set of permutation matrices for each $l$ value, so that the datasets with the same $l$ are exactly the same for different methods.

---

### Review · Reviewer_gRVi · 2024-04-30

**Summary Of Contributions:**

⁤This paper aims to develop a graph generation model that addresses the limitations of permutation-invariant diffusion models. ⁤⁤Initially, the authors analyze the performance of existing invariant models. ⁤⁤They identify a degradation in performance due to restrictive architectures and an increased number of target distribution modes. ⁤⁤This analysis highlights the need for a more flexible approach. ⁤⁤Subsequently, the authors introduce SwinGNN, a GNN that maintains permutation-invariant sampling without being invariant and utilizes shifted-window self-attention to enhance scalability for large graphs. ⁤⁤Empirical results demonstrate SwinGNN's superiority over existing methods across synthetic and real-world datasets.

**Audience:**

Yes

**Claims And Evidence:**

Yes

**Requested Changes:**

I'm intrigued by how the theories presented in this paper might apply to graphs constructed using radial cutoffs. Unlike graphs generated by sampling adjacency matrices, these graphs are created by sampling node coordinates and constructing adjacency matrices based on pairwise distances between sampled nodes. Does the theory in this paper suggest that we need to reconsider the design of such architectures too? Alternatively, could the paper offer a clear explanation of the scope of diffusion models, specifically outlining the implications of the theory it covers?

**Strengths And Weaknesses:**

**Strengths:**
1. The paper offers valuable insights, both theoretical and empirical, into the potential limitations of permutation-invariant models and the necessity for invariant sampling. These findings hold significant importance for future design considerations.
2. Building upon these insights, the authors introduce SwinGNN, a non-invariant framework designed to enhance performance and efficiency in capturing intricate real-world graph distributions while ensuring sampling invariance.

---

> ### Author Response · Authors · 2024-05-12
>
> > $\\textbf{Q1@gRVi}$: I'm intrigued by how the theories presented in this paper might apply to graphs constructed using radial cutoffs. Unlike graphs generated by sampling adjacency matrices, these graphs are created by sampling node coordinates and constructing adjacency matrices based on pairwise distances between sampled nodes. Does the theory in this paper suggest that we need to reconsider the design of such architectures too?
>
> $\\textbf{A1@gRVi}$: Thank you for the insightful comment. We believe that our theoretical analysis on permutation also applies to radial cutoff graph construction based on node coordinate generation.
>
> Specifically, let $X=[X\\_1, X\\_2, \\cdots, X\\_n] \\in \\mathbb{R}^{n \\times d}$ represent the node coordinates, which are sampled from some generative models. Let $Y = g(X) \\in  \\mathbb{R}^{n \\times n}$ denote the constructed graph adjacency matrix, where $g$ is the construction algorithm based on pairwise distances, which is a deterministic function and considered to be a post-processing step. Our analysis and conclusion mainly affects the first stage of node coordinate generation.
>
> First, our theoretical analysis of the number of modes in target distributions (Lemma 4.1) is still applicable. For $n$ nodes' coordinate features, there are $O(n!)$ equivalent but different matrix representations (each in shape of $\\mathbb{R}^{n \\times d}$). Namely, a single graph's node features would incur $O(n!)$ modes in its permutation-invariant target distribution. Assuming a dataset with $m$ such data points, each consisting of $n$ nodes, the number of modes in its target distribution would increase to $O(n!m)$. This would make the learning more difficult as in our case of adjacency matrix generation.
>
> Secondly, we conduct experiments on molecule generation, where the node features are jointly generated along with the adjacency matrix. The joint generation objective is arguably more challenging than node feature generations alone.  Experimental results in Table 3 show our method outperforms the current baselines that use equivariant architectures. This implies that the same design principle may also boost performance for node coordinate feature generation alone.
>
> We thank the reviewer for this constructive comment, and we look forward to investigating this problem in the future.
>
>
> > $\\textbf{Q2@gRVi}$: Alternatively, could the paper offer a clear explanation of the scope of diffusion models, specifically outlining the implications of the theory it covers?
>
> $\\textbf{A2@gRVi}$: Thanks for your comment. We believe that our theoretical analysis applies to generative models aimed at learning permutation-invariant target distributions in general. These distributions may involve specific datasets such as node features (order-1 tensor), adjacency matrices (order-2 tensor), or arbitrarily higher-order graph data (order-$k$ tensor). Interestingly, recent work AlphaFold3 [1] also empirically verifies that diffusion generative models applied to molecule/graph data do not necessarily require an equivariance structural bias.
>
> \
> [1]: Abramson, J., Adler, J., Dunger, J., Evans, R., Green, T., Pritzel, A., Ronneberger, O., Willmore, L., Ballard, A.J., Bambrick, J. and Bodenstein, S.W., 2024. Accurate structure prediction of biomolecular interactions with AlphaFold 3. Nature, pp.1-3.

---

### Author Response · Authors · 2024-05-12

Dear reviewers,

Thank you for your insightful comments. In the response below, we have provided further explanations to your comments and questions. We have also modified the manuscript (marked in blue text) to better address the comments.

We apologize for the LaTeX display error, which seems to be due to some bugs in the OpenReview system; the preview on our end looks fine. Please do not hesitate to let us know if you have any further questions.

---

### Author Response · Authors · 2024-06-11
**Follow-Up on Rebuttal and Discussion**

Dear Reviewers and Action Editor,

I hope this message finds you well.

It has been about four weeks since we submitted our rebuttal to the reviewers' comments. We would be happy to provide further clarification on any questions if needed. We look forward to discussing with you.

Best regards,
The Authors

---

### Decision · Action_Editor_MP3R · 2024-06-12

**Recommendation:** Accept as is

**Comment:**

The three reviewers agree that the paper is relevant to the TMLR community and make substantial substantial contributions that may prove useful for future design considerations within diffusion models. The round of reviews and revisions made to the paper improved its quality and clarity.

**Audience:**

Yes.

**Claims And Evidence:**

Yes.